# Dissecting transcriptional amplification by MYC

Zuqin Nie[1†], Chunhua Guo[2†], Subhendu K Das[1], Carson C Chow[3], Eric Batchelor[1,4,5]*, S Stoney Simons Jnr[2]*, David Levens[1]*

[1]Laboratory of Pathology, CCR, NCI, NIH, Bethesda, United States; [2]Steroid Hormones Section, NIDDK/LERB, NIH, Bethesda, United States; [3]Mathematical Biology Section, NIDDK/LBM, NIH, Bethesda, United States; [4]Laboratory of Cell Biology, CCR, NCI, NIH, Bethesda, United States; [5]Department of Integrative Biology and Physiology, University of Minnesota, Minneapolis, United States

**Abstract** Supraphysiological *MYC* levels are oncogenic. Originally considered a typical transcription factor recruited to E-boxes (CACGTG), another theory posits MYC a global amplifier increasing output at all active promoters. Both models rest on large-scale genome-wide "-omics'. Because the assumptions, statistical parameter and model choice dictates the '-omic' results, whether MYC is a general or specific transcription factor remains controversial. Therefore, an orthogonal series of experiments interrogated MYC's effect on the expression of synthetic reporters. Dose-dependently, MYC increased output at minimal promoters with or without an E-box. Driving minimal promoters with exogenous (glucocorticoid receptor) or synthetic transcription factors made expression more MYC-responsive, effectively increasing MYC-amplifier gain. Mutations of conserved MYC-Box regions I and II impaired amplification, whereas MYC-box III mutations delivered higher reporter output indicating that MBIII limits over-amplification. Kinetic theory and experiments indicate that MYC activates at least two steps in the transcription-cycle to explain the non-linear amplification of transcription that is essential for global, supraphysiological transcription in cancer.

*For correspondence:
ebatchel@umn.edu (EB);
stoney.simons@gmail.com (SSSJ);
levensd@mail.nih.gov (DL)

[†]These authors contributed equally to this work

**Competing interests:** The authors declare that no competing interests exist.

## Introduction

The *MYC* proto-oncogene drives the pathogenesis of most human malignancy (*Dang, 2012*). *MYC* encodes a 439 residue nuclear bHLH-ZIP protein that, when expressed at supraphysiological levels, activates virtually all the hallmarks of cancer (*Hanahan and Weinberg, 2011*). Upon dimerization with MAX, MYC becomes a DNA binding protein with a preference, but not absolute specificity, for binding with E-boxes (CACGTG) (*Blackwood and Eisenman, 1991*; *Guo et al., 2014*). Although originally construed to be a master transcription factor dictating critical cellular decisions via E-boxes at strategic targets, an alternate hypothesis posits MYC to be a global amplifier, binding and increasing expression at all already active promoters (*Lin et al., 2012*; *Nie et al., 2012*). According to this explanation, MYC-amplifier-gain increases as the level of signaling to a transcriptionally activated promoter increases. Thus, highly expressed genes are disproportionately amplified compared with less expressed genes. Both the global amplifier and gene selective activator models for MYC action have been buttressed by large-scale genome-wide expression and ChIP-Seq studies—sometimes sustaining opposite conclusions from the same data (*Kress et al., 2015*; *Lorenzin et al., 2016*; *Sabò et al., 2014*; *Tesi et al., 2019*; *Wolf et al., 2015*). Because the selection of statistical parameters and models, as well as the choice of computational algorithms, determines their outputs, these '-omics' studies have failed to resolve the essential role of MYC in transcription.

MYC interacts with a wide assortment of transcription and chromatin components as well as with other proteins both in vitro and in vivo (*Agrawal et al., 2010*; *Baluapuri et al., 2019*; *Büchel et al.,*

*2017*; *Chakravorty et al., 2017*; *Chan et al., 2014*; *Dingar et al., 2015*; *Kalkat et al., 2018*; *Koch et al., 2007*). Whether MYC exploits distinct complexes to differently control the transcription of diverse targets, or whether it acts stereotypically at all active genes where expression may also be modified according to the local chromatin landscapes, is not known.

At natural promoters, in vivo, the essential function of MYC may be masked or modified by other transcription and chromatin components. The genomic landscapes of most genes are studded with *cis*-elements that recruit a plethora of specific and general transcription factors and repressors, as well as chromatin remodeling and modifying complexes, all of which may complicate or camouflage the fundamental activity of MYC. The present study was initiated to isolate MYC's role in gene expression away from confounding influences. We have used classical and synthetic biology strategies to interrogate directly MYC's ability to drive reporter gene expression in vivo rather than inferring MYC function from 'omics' studies.

First MYC's effect on global RNA production was monitored in a tightly regulated system. Next we sought to express MYC from transiently transfected or inducible integrated lentiviruses to test its effect on the output of minimal-promoter-driven reporters, either via transient transfection or chromosomally integrated lentivirus. MYC-driven promoters were tested to see if transcription was E-box-dependent versus E-box boosted as suggested by *Nie et al., 2012*. The intrinsic activation by MYC versus amplifier action in the absence or presence of exogenously supplied activators, was measured, respectively. The contributions of the conserved protein motifs (MYC boxes, MBs) in the MYC activation domain (MYC boxes, MBI-IV [*Conacci-Sorrell et al., 2014*; *Tansey, 2014*]) were assessed. In design, execution and interpretation, these synthetic biology experiments are completely orthogonal to and independent from the genomic studies that provoked the amplifier hypothesis. Yet the reporter responses to MYC and to the heterologous transcription activators employed in these experiments are entirely coherent with the general amplifier model. Moreover, kinetic studies of synthetic biology experiments on a steroid-induced gene (*Blackford et al., 2014*; *Chow et al., 2011*; *Dougherty et al., 2012*) identified the mode of action of MYC in this system as being either a facilitator of a transcription factor or a transcriptional accelerator that binds weakly in at least two locations after receptor-steroid binding. Finally, the general amplifier model provides a framework to understand the properties of the MB mutations. The most straightforward interpretation of these experiments is that MYC is a general amplifier that acts preferentially on highly expressed genes.

## Results

### MYC rapidly increases total cellular RNA

Whether MYC immediately and directly leads to a global increase in RNA levels has been a matter of some controversy. Any attempt to assess the effect of MYC on RNA levels across all genes individually would be highly dependent on the particular spike-in methods and computational algorithms used to quantify the results. It would also depend on the kinetics of whatever system is employed to acutely activate or inactivate MYC and on the basal level of MYC in cells studied. Because total RNA is 97% rRNA, the global increase in RNA is primarily due to increased rRNA synthesis. Since total RNA and total poly-A RNA generally move in parallel, total RNA synthesis is often a reasonable proxy for mRNA synthesis. Surprisingly, *Tesi et al., 2019* saw no increase in global RNA transcription until 24 hr post-activation, using an in situ assay, and concluded that MYC increased total RNA (and hence rRNA) indirectly. Yet ribosomal proteins and rRNA have long been recognized as direct MYC targets from flies (*Grewal et al., 2005*) to humans (*Gomez-Roman et al., 2003*; *Grandori et al., 2005*). We found an early increase in total RNA using HO15.19 MYC-ER cells (*O'Connell et al., 2003*). These MYC-knockout rat cells stably express tamoxifen-activated MYC-ER. Cells were serum starved for five days and then treated with 200 nM of tamoxifen. Previous work demonstrated an immediate increase in rRNA transcription following tamoxifen-treatment of these cells (*Schlosser et al., 2003*). From an already large baseline, total RNA increased to near statistical significance at 6 hr and became so, after 10 and 14 hr (*Figure 1—figure supplement 1*). Such an experimentally detectable increase in total RNA must largely include increased rRNA, snoRNAs, 5S rRNA, and the host of mRNAs that are required for ribosome biogenesis.

## A synthetic, minimal system to study MYC amplifier activity

To interrogate MYC function while avoiding confounding genomic influences, MYC-expression and reporter plasmids were co-transfected into U2OS osteosarcoma cells (*Figure 1A*). (U2OS cells have been employed extensively for studies of MYC target genes [*Lorenzin et al., 2016*; *Walz et al., 2014*]). The expressed MYC was a fully functional *MYC-EGFP* fusion gene (*Nie et al., 2012*) driven by the CMV early promoter and enhancer. This fusion protein, when expressed homozygously at the *Myc* locus in mice, dimerizes with *Max* and supports normal development and physiology (*Nie et al., 2012*). Immunoblot of transfected cells confirmed that the output of MYC protein was linearly related to the transfected amount of the *MYC*-encoding plasmid (*Figure 1B*). (Note that under the immunodetection conditions shown here, endogenous MYC was not seen, and that transfected MYC was not detected with less than 10 ng of transfected MYC cDNA plasmid.) The co-transfected reporter plasmids expressed firefly luciferase (luc) or renilla luciferase (ren) from a Glucocorticoid Response-Element (*GRE*)-driven minimal herpesvirus (HSV) thymidine kinase (tk) or thromboxane synthase (ts) promoter, respectively (*Figure 1A* and *Figure 1—figure supplement 2*). *tsRen* was included as a control in all transient transfection experiments. (The ts promoter supports such low levels of expression that it would be expected to be barely amplified by MYC [*Nie et al., 2012*]. Because U2OS cells contain almost no glucocorticoid receptor (GR) (*Blackford et al., 2012*; *Lee and Simons, 2011*; *Rogatsky et al., 1997*) a small amount of GR was supplied by transfection. GRE-activity in these cells has been well-characterized and is completely dependent on the addition of exogenous glucocorticoids (*Blackford et al., 2012*; *Chow et al., 2011*; *Dougherty et al., 2012*; *Lee and Simons, 2011*; *Rogatsky et al., 1997*; *Tao et al., 2008*). Besides the minimal HSV *tk* promoter, another variant of the *tkLuc* reporters also included an E-box as well as a GRE (see below).

## Non-linear transfer function for MYC amplification

According to the amplifier hypothesis, MYC-driven output of active genes intensifies as intrinsic unamplified promoter strength increases and, at promoters bearing E-boxes, the output may be further boosted. Accordingly, the weak outputs of *tkLuc*, GRE*tkLuc* and *tsRen* were only marginally and comparably augmented by co-transfected MYC in the absence of dexamethasone (DEX) (*Figure 1C*). Upon treatment with DEX and *MYC*, *GREtkLuc* increased ~15 fold, whereas the other reporters remained unresponsive. Therefore, upon activation by GR and DEX, *GREtkLuc* became more MYC-responsive just as predicted by the amplifier theory (*Figure 1C*). Consistent with this theory, the amount of additional transactivation delivered by added MYC was larger when more GR was used to cause greater activation of the *GREtkLuc* reporter (*Figure 1—figure supplement 3A*). As the level of transfected MYC-EGFP was progressively increased, the reporter response increased non-linearly as if MYC activation enhanced the efficiency of further MYC activation (*Figure 1C*).

At high concentrations, MYC also weakly amplified the output of *GREtkLuc* when activated by co-transfected progesterone receptor (PR-B) along with the progestin R5020 (*Figure 1—figure supplement 3B*). The reduced gain of the amplifier in this case was consistent with weaker activation by PR relative to GR and is explained by the non-linear transfer function (input-output curve according to MYC dose) that has been suggested to be a defining characteristic of MYC amplification (*Lorenzin et al., 2016*; *Nie et al., 2012*; *Wolf et al., 2015*).

## MYC amplification requires MYC-MAX dimerization

Most studies of MYC's action have indicated that it must dimerize with MAX to exert its influence on gene expression. MYC does not bind to DNA alone. To test if transcription amplification was dependent upon MYC-MAX dimerization, MYC—amplified GR/DEX-driven *GREtkLuc* expression was challenged with 10058-F4 (F4) (*Figure 1D*). This well-characterized inhibitor of MYC-MAX dimerization (*Wang et al., 2007*) not only dramatically impaired amplification, but unexpectedly almost abolished the DEX-dependent activation of *GREtkLuc* by GR in U2OS cells that express only endogenous MYC (*Figure 1C vs. D*, red bracket). This result indicates that the activation of the *GREtk*-promoter by Dex in U2OS cells is dependent on MYC-Max dimers, whether endogenous or transfected.

## E-boxes augment, but do not define MYC action at reporter promoters

Does the transcriptional response of MYC target genes demand MYC binding at E-boxes, or is the response graded upwards according to the real, but modest E-box binding selectivity of MYC-MAX?

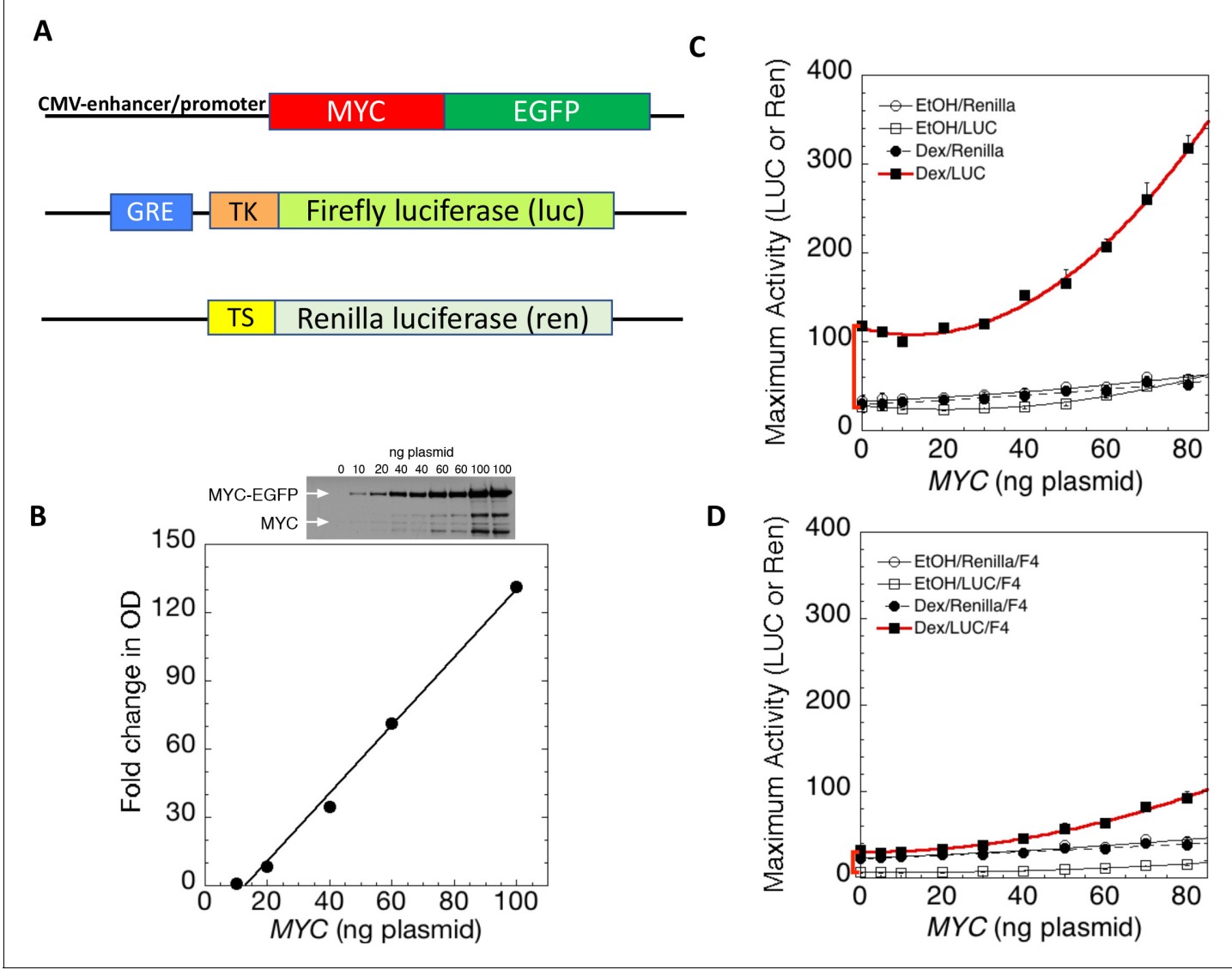

**Figure 1.** MYC amplifies glucocorticoid receptor-primed gene expression. (**A**) Cells were transfected with plasmids (top to bottom) expressing MYC-EGFP driven by the CMV-enhancer promoter, firefly luciferase (luc) driven by a glucocorticoid response element (GRE) from a Herpes thymidine kinase (TK) minimal promoter, and renilla luciferase (ren) driven by the basal thromboxane synthase promoter. (**B**) MYC-EGFP protein expression parallels the transfected amount of MYC-EGFP plasmid. Note that the linear fit of immunoreactive MYC vs. transfected plasmid indicates that negligible MYC is expressed below 13 ng of plasmid. The endogenous MYC is calculated to equal ≈2.5 ng of MYC plasmid. Therefore, zero total MYC is calculated (*Chow and Simons, 2018*) to be ≈10 ng of plasmid (=13–2.5). Top: Immunoblot of cells transfected with different amounts of *MYC-EGFP* plasmid. (**C** and **D**). Total Renilla or Luciferase activity with EtOH or high Dex concentrations (50 or 100 nM) in U2OS cells transiently transfected with 2 ng of GR plasmid, 100 ng GREtkLUC reporter, and the indicated amounts of MYC plasmid, without (**C**) or with (**D**) 50 µM F4 were determined as described in Materials and methods, averaged, and plotted ± SEM (n = 2 or 4). $R^2$ values for polynomial curve fits were all ≥0.96. Note that GR-induction using endogenous MYC is sensitive to F4 (red brackets on Y-axes of C and D).

The online version of this article includes the following figure supplement(s) for figure 1:

**Figure supplement 1.** MYC acutely increases cellular RNA levels.

**Figure supplement 2.** E-box-like sequences and base composition of transient and stable reporters.

**Figure supplement 3.** MYC amplification of steroid receptor activity.

Comparison of the expression and binding of MYC at genomic promoters with or without E-boxes has indicated that E-boxes augment, but are not required for MYC-responsiveness (*Nie et al., 2012*). The affinity of MYC-MAX for the best binding of all possible eight-mer base sequences exceed those of the worst by less than 100-fold (modest for sequence specific recognition)

(*Guo et al., 2014*). So, one perfect E-box has the same DNA binding avidity as 100 bp of non-E-box DNA, and the insertion of a perfect E-box into 200 bp of E-box-less promoter DNA would then be expected to increase the region's net avidity for MYC by ≈ 1.5 fold (*Figure 2—figure supplement 1*). If activity parallels binding, then E-box reporter output should exceed non-E box reporter output by this same degree. To test this prediction, an E-box was inserted into *GREtkLuc* yielding *GRE-E-box-tkLuc* (*Figure 2A*). In the absence of Dex, the low output of *GRE-E-box-tkLuc* was only marginally more responsive to exogenous MYC (*Figure 2B*) than *GREtkLuc.* Addition of Dex licensed 1.5x greater MYC amplification of *GRE-E-box-tkLuc* than of *GREtkLuc* by exogenous MYC, exactly as predicted (*Figure 2B vs. C*). This single E-box was insufficient to drive switch-like, on-off high-level reporter expression. Notably, though *GRE-E-box-tkLuc* initially supported lower reporter expression than did *GREtkLuc,* higher MYC levels empowered greater E-box-driven output. At low levels of intracellular MYC, MAX-MAX dimers prevail and may bind to E-boxes more specifically, but less avidly than the more sparse MYC-MAX (*Conacci-Sorrell et al., 2014*; *Guo et al., 2014*). MAX-MAX dimers possess intrinsic repressor activity (*Conacci-Sorrell et al., 2014*), and MAX heterodimerizes with other repressive bHLH-ZIP proteins (*Conacci-Sorrell et al., 2014*). As MYC-MAX dimerization swells, amplification through E-boxes prevails.

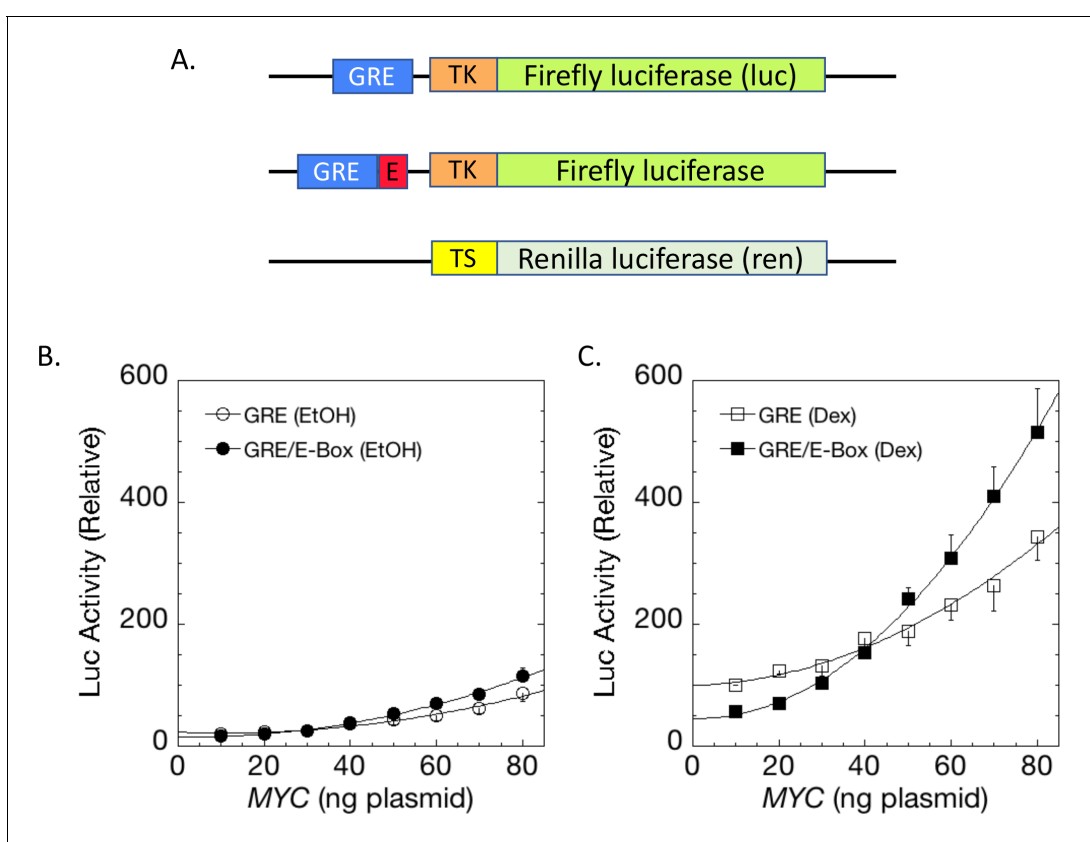

**Figure 2.** E-boxes assist but are not essential for MYC amplifier activity. (**A**) Dex-inducible reporter plasmids without or with E-boxes (top and middle, respectively) were co-transfected with a basal and uninducible Renilla-expressing reporter (bottom). (**B**) Without Dex induction, MYC marginally influences reporter expression with or without E-boxes. (**C**). With Dex, E-boxes depressed and then augmented reporter expression at low and high levels of MYC, respectively. Graphs are averages of 4 paired experiments ± S.E.M. in U2OS cells transiently transfected with 2 ng of GR plasmid, 100 ng GREtkLUC reporter without or with an E-box. All data were normalized to the value for 10 ng *MYC* with Dex on GREtkLUC reporter before averaging ($R^2$ values for polynomial curve fits are ≥0.98).

The online version of this article includes the following figure supplement(s) for figure 2:

**Figure supplement 1.** MYC operates most effectively when bound within a couple of hundred bases of a transcription start site (*Nie et al., 2012*).

## Participation of conserved MYC box (MB) motifs in transcription amplification

The evolutionarily conserved sequence motifs MBI-IV had been presumed, and in many instances shown, to mediate MYC biological activity and contribute to oncogenesis. To ascertain whether and which MBs contribute to amplifier activity, amino acid substitution mutations were engineered into conserved segments of the MBs within MYC-EGFP (*Figure 3A*) and assessed for function by co-transfection with a reporter - *GREtkLuc* and GR as well as with *TS-Ren*. Although mutation of no individual MB abrogated amplification compared with wild-type MYC (*Figure 3B–F*), substitutions within MBII considerably attenuated amplifier gain (*Figure 3D*).

Strikingly, at high levels of transfected MYC, substitution mutations within MBIII dramatically boosted the amplifier gain on GR activated reporter expression (*Figure 3E*). MBIV substitutions also increased MYC amplification (*Figure 3F*). Over-amplification by the mutant MBs III and IV was attenuated by co-mutation of MBI (*Figure 3G and H*). Thus, MBIII and IV mutations augment the dynamic range of MYC amplification but are not dominant over the inhibition by the MBI mutation. Mutation of MBIII also enhanced MYC amplification of PR-B (*Figure 1—figure supplement 3B and C*), and this enhancement was also markedly attenuated by co-mutation of MBI (*Figure 1—figure supplement 3C*).

Mutant MYC protein levels were checked by immunoblot to rule out simple protein dose explanations. Mutant MBIII-hyperactivation does not result from increased levels of expressed protein. In fact, MBIII-mutant protein levels were markedly reduced compared to wild-type MYC (*Figure 3I*; it should be noted that the rabbit monoclonal antibody used to detect MYC binds with epitopes far-removed to the amino-terminal-side of MBIII and IV and that immunoblots of wild-type and mutant MYC-EGFP fusion proteins using either anti-MYC or anti-EGFP are equivalent, *vide infra*).

While MBI mutations have been implicated in the stabilization of MYC protein, ubiquitylation and turnover of MYC have been associated with increased transactivation . To visualize the relationship between MYC transactivation and protein levels for wild-type and mutant MYC proteins, reporter activity versus MYC proteins levels (as assessed by immunoblot) were plotted (*Figure 3J*). A striking inverse relationship was apparent between activity and MYC levels with the highly active MBIII and MBIV mutants sustaining the least amount of MYC, whereas minimally active double and triple MB-mutants displayed the highest levels of MYC protein. This inverse relationship between abundance and activity is consistent with a previous suggestion that transactivation by MYC and its turnover are coupled (*Farrell et al., 2013*; *Farrell and Sears, 2014*; *Kim et al., 2003*; *Molinari et al., 1999*; *Salghetti et al., 2000*).

## MYC-Box mutations change both non-E and E-box promoter-output at the RNA level

Though MYC augmented GR-activation of reporters with and without E-boxes, formally the possibility remained that E-box-dependent and independent, upregulated expression might occur via different mechanisms. For example, global increases in cellular macromolecular synthetic pathways, such as translation, might provide an indirect component to amplification distinct from specific transcriptional upregulation. The mutations in MBII and MBIII were compared with wtMYC for transcriptional activation/amplification of reporters with and without E-boxes (*GRE-Ebox-tkLuc and GREtkLuc*). With high levels (80 ng) of transfected *MYC-EGFP* plasmids, the reporter activity profiles were similarly shaped with and without E-boxes, though inclusion of an E-box boosted expression by ~1.5 fold (*Figure 4A*). Just as in *Figure 3*, MBII decreased expression whereas MBIII increased expression (*Figure 4A*). wtMYC and mutant-MBIII were transfected along with *GREtkLuc* and reporter expression, normalized to co-transfected *tsRen*, was compared at the level of luciferase activity (*Figure 4B*) and RNA (luciferase-RTqPCR, *Figure 4C*). The transfected MYC increased reporter RNA levels. The background RNA seen with the empty vector likely represents random transcription extending throughout the plasmid backbone. This is a long recognized problem for quantifying plasmid-encoded mRNAs recovered from transfected cells. (*Boshart et al., 1992*; *Groskreutz and Schenborn, 1997*). Thus, MYC mutations do not discriminate activation of expression genes with or without E-boxes.

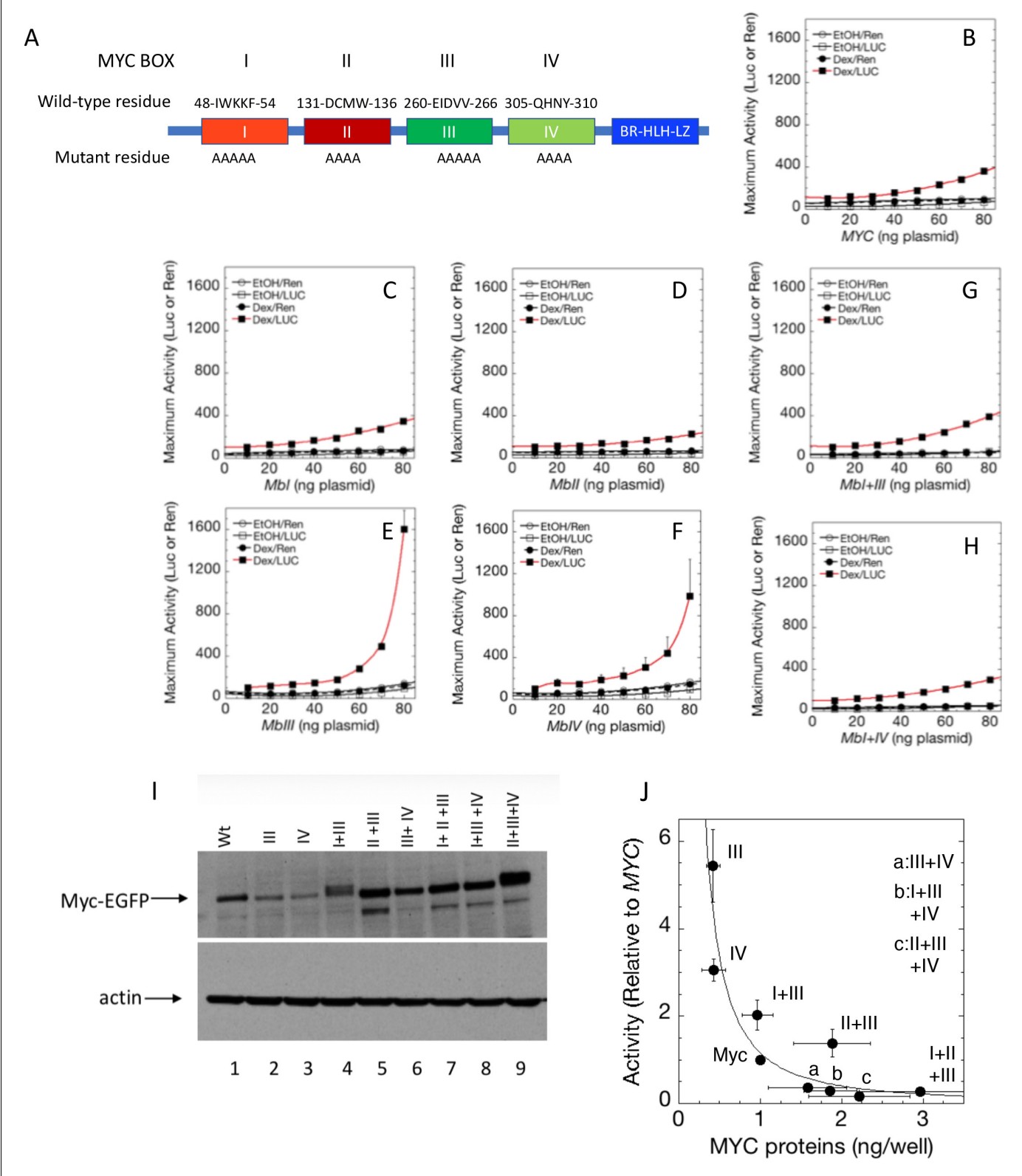

**Figure 3.** Cooperation between Myc Boxes (MBs) is required for amplifier activity. (**A**) Summary of mutations introduced into the MBs of transfected *MYC*. B-H. Effect of various mutations in MBs, individually or in combination, on MYC amplifier activity. Total Renilla or Luciferase activity with EtOH or high Dex concentrations (50 or 100 nM) in U2OS cells, transiently transfected with 2 ng of GR plasmid, 100 ng GREtkLUC reporter, and the indicated amounts of MYC plasmid, were determined as described in Materials and methods, normalized to that for 10 ng plasmid, averaged, and plotted ± SEM

*Figure 3 continued on next page*

*Figure 3 continued*

(n = 2, 3, or 4) for **B**) wild-type, (**C**) MBI mutation, (**D**) MBII mutation, (**E**) MBIII mutation, (**F**) MBIV mutation, (**G**) MBI+III mutation, and (**H**) MBI+IV mutation. $R^2$ values for polynomial fits of Dex/Luc curves were all ≥0.96 except for E and F, which were fit to smooth curves. (**I**). MYC protein levels (immunoblot) of transfected wild-type (WT) and mutant plasmids. (**J**) Inverse relationship between MYC protein level and amplifier activity for wild-type and mutant proteins. Data are averages ± S.E.M. from 2 to 5 experiments (n = 9 for wt MYC).

## MYC amplifies the output of synthetic transcription activators

Although MYC stimulation of GR-activated transcription seemed exactly as predicted by the general amplifier hypothesis, it might alternatively reflect specific transcription factor synergy between MYC and GR (or PR), perhaps as part of a selective physiological response. To discriminate whether MYC-augmented reporter expression represented specific synergy with GR versus general amplification of transcription, assays were run to test if reporters controlled by synthetic activators could be driven by MYC to levels beyond those achievable by such chimeric activators alone. These experiments used the well-characterized chimeric GAL4/VP16 in which the DNA binding domain (DBD) of the yeast activator GAL4 is fused with an activating segment of the Herpes simplex ICP4 gene (*Figure 5A*; *Sadowski et al., 1988*). The GAL4 DBD self-dimerizes to bind the 17 bp *GAL4* Upstream Activating Sequence (UAS). Importantly the human genome harbors no orthologous sequences for either the Gal4 DBD or the VP16 activating segment. Therefore, U2OS cells are unable to drive Gal4 UAS-reporters unless supplied with exogenous *GAL4/VP16*. As is typical, five tandem *Gal4UAS*s were inserted upstream of a TATA-box (5x*GAL4UAS-TATALuc*) to maximize activation by GAL4/VP16. Alone, Gal4UAS-GAL4/VP16-driven luciferase expression saturated at a low dose (2 ng/well) and then declined slowly (*Figure 5B*). Unactivated renilla activity declined slightly faster. Both responses are likely due to squelching (*Gill and Ptashne, 1988*), which is when the transcriptional activity of a factor A binding with a target gene is inhibited by another transcription factor B—even when B does not bind to the target. This can occur in multiple scenarios. For example, if factor A interacts independently with both cofactors B and C but is active only within a complex of A+B+C, then excess A will form AB and AC but little ABC, in which case transcriptional activity with the target gene will first increase and then decrease as more A is added. Co-transfected MYC progressively amplified GAL4/VP16-driven reporter activity (*Figure 5C*), just as it had for GR. The output of Gal4/VP16 and MYC together exceeded those of either transactivator alone, indicating non-redundant mechanisms. These results support the notion that MYC is indeed a general amplifier of activated transcription.

MYC amplification of GAL4/VP16 activation was tested for the involvement of MYC boxes. Similar to GR and PR, mutation of MBIII dramatically raised the gain on amplification of GAL4/VP16 at high levels of MYC (*Figure 5D*). This result suggests that MBIII helps to limit positive feedback of MYC on VP16-driven expression. Likewise, co-mutation of MBI and MBIII reduced the excessive gain of the MBIII mutation back to wild-type levels (*Figure 5E*), inviting consideration of MBI as a regulator of mutant MBIII activity.

## MYC amplifies expression from chromosomally integrated minimal promoters

The experiments above suggest that MYC amplifies transcription on transfected DNA templates. Does the MYC-amplifier also operate on chromosomally embedded, properly replicating minimal promoters? To address this question, U2OS cells were infected with a panel of integrating lentiviruses that delivered E1b-minimal promoters driving the expression of mCherry red fluorescent protein (*CFP*). These chromosomally integrated reporters carried a single *GAL4UAS* upstream of the *CFP* transcription start site, and alternative reporter variants either including a single E-box (*GAL4UAS/E*) or not (*GAL4UAS*) (*Figure 6A*). Host cells also harbored a lentivirus that expressed from a tetracycline inducible promoter the same MYC-EGFP fusion protein used for the above transfection experiments (*Figure 6A*). Thus, in the absence of induction, reporters were exposed only to endogenous U2OS MYC, whereas upon induction with doxycycline, the reporter promoters were bathed with increased levels of MYC. Utilization of a single *GAL4UAS* assured that reporter expression was confined within bounds typical of endogenous genes. When included, *GAL4/VP16* was co-

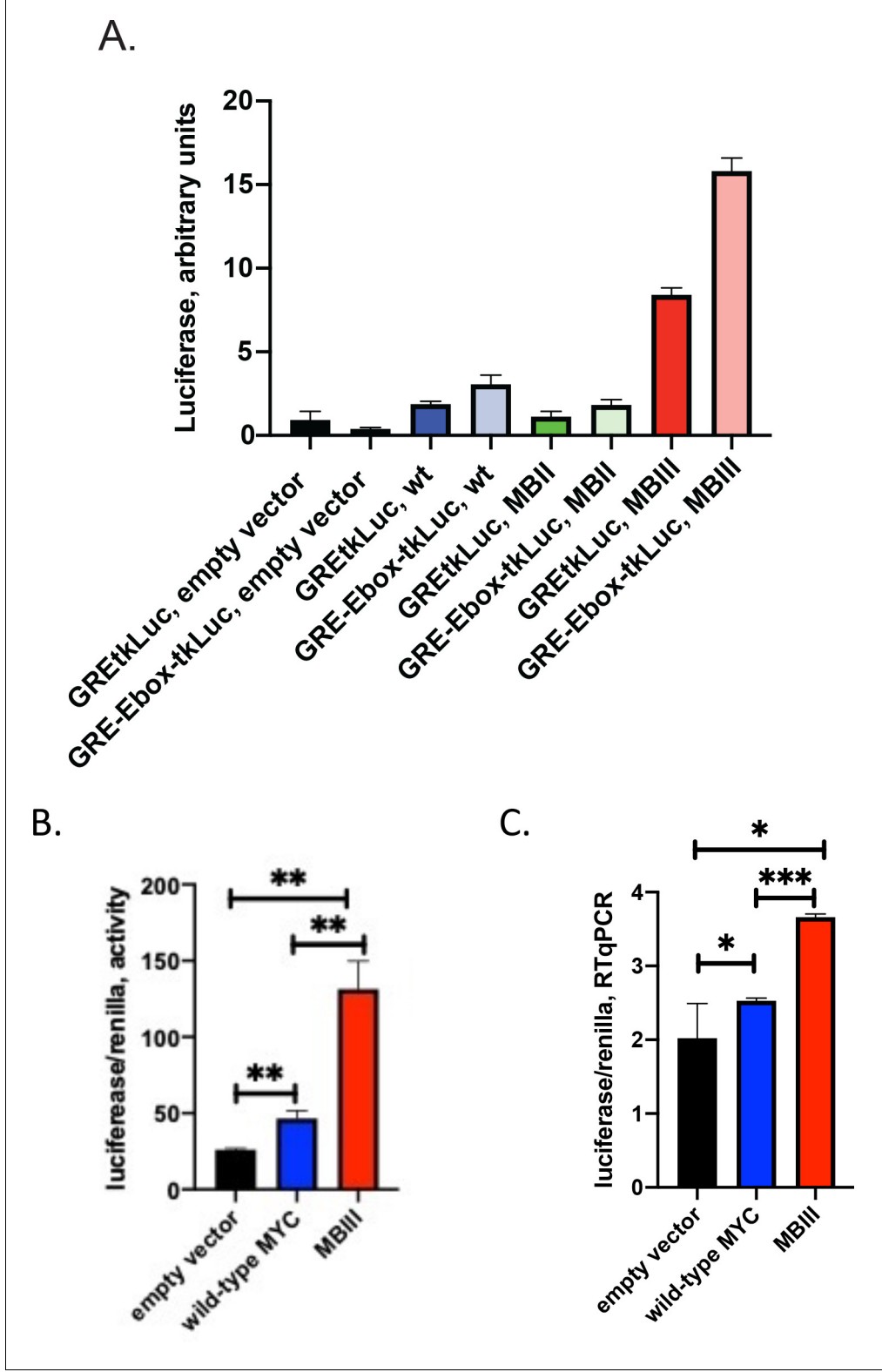

**Figure 4.** MYC-Box mutations change both non-E and E-box promoter-output at the RNA level. (**A and B**) Cells were transfected as noted with 80 ng of empty vector, MYC, MBII or MBIII expressing vectors, 100 ng of non-E-box or E-box reporter and 2 ng of GR plasmids . Luciferase and renilla luciferase were assayed. (**C**). Transfected cells as in B were harvested and RNA extracted for RT-qPCR using Luna Universal One-Step RT-qPCR Kit (New England Biolabs). Experiments were performed in biological triplicate. Standard deviations are indicated. *,** and

*Figure 4 continued on next page*

*Figure 4 continued*

\*\*\* indicate p≤0.05, 0.01, 0.001, respectively, one-tailed t-test. Experiments performed in triplicate (A and B, n = 3; C, n = 1).

transfected with a plasmid expressing monomeric palmitoyl-mTurquoise to visually mark transfected cells.

Alone, endogenous levels of U2OS MYC sustained only low basal mCherry expression from the E-box-less minimal *GAL4UAS* promoter (*Figure 6B*). Induction of MYC with added Dox or separately transfecting GAL4/VP16, each failed to augment reporter output (upper right and lower left panels of *Figure 6B*); however together, transfected *GAL4/VP16* and induced-MYC cooperated to raise chromosomal report output (lower right panel of *Figure 6B*).

Including an E-box along with *GAL4/UAS* proved insufficient to increase reporter levels when driven only by basal levels of endogenous MYC, with or without GAL4/VP16 (upper panels of *Figure 6C*). Upon MYC induction with Dox, E-box-bearing reporter output intensified and achieved its highest levels when *GAL4/VP16* was co-transfected (lower right panel of *Figure 6C*). These data anticipate that MYC binds at both E-box-less and E-box-bearing promoters (*Guo et al., 2014*; *Nie et al., 2012*), but a bit more efficiently at the latter in accord with the amplifier hypothesis. As with transient transfection, MYC-amplification to mid-level expression was resilient to mutation of

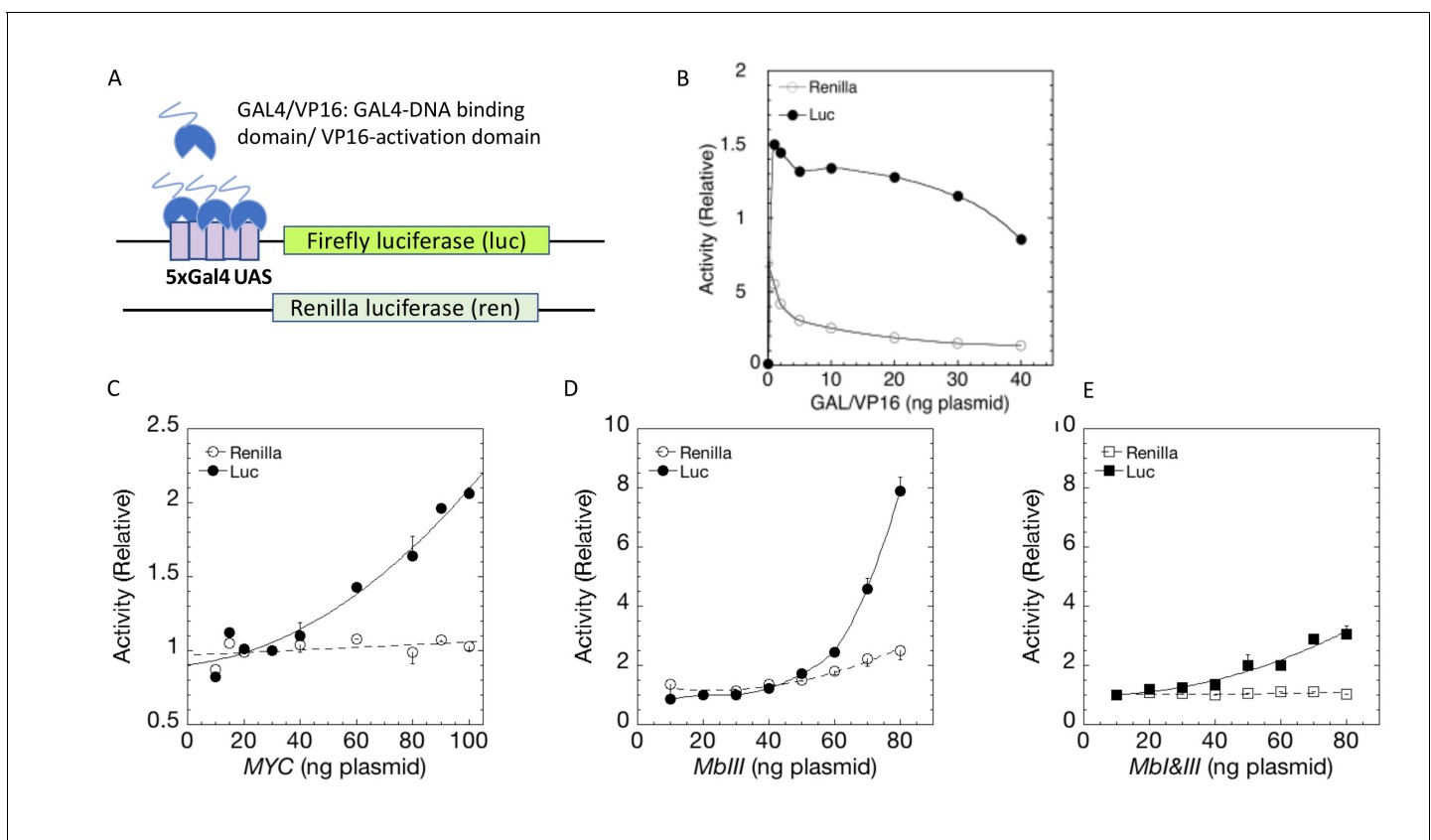

**Figure 5.** MYC amplifies synthetic transcription factor GAL4/VP16 output. (A) Cells were co-transfected with varying amounts of plasmids expressing the GAL4/VP16 chimeric protein that includes the DNA binding domain of GAL4 and the transactivation domain of VP16, as well as Gal4UAS-luc and *tsRen*. (B) GAL4/VP16 transactivation of Gal4UAS-luc (10 ng) saturates at ≈2 ng of transfected *GAL4/VP16*. Renilla TS was present at 10 ng. Activity is relative to 10 ng *GAL4/VP16*. Data points are averages of triplicate determinations ± SD of one experiment. (C) MYC amplifies GAL4/VP16 output. (D) Mutation of MBIII unrestrains the gain of MYC amplification of GAL4/VP16. (E). Unrestrained amplification of GAL4/VP16 activation by MBIII-mutant MYC requires intact MBI. In C-D, total amount of *MYC* vector was kept constant by addition of decreasing amounts of pd4EGFP plasmid. Data were normalized to that for 30 ng (C), 20 ng (D), or 10 ng (E) of *MYC* construct plasmid, each with 10 ng each of GAL4/VP16 and of GAL4UAS-luc plasmid, averaged, and plotted ±range (n = 2).

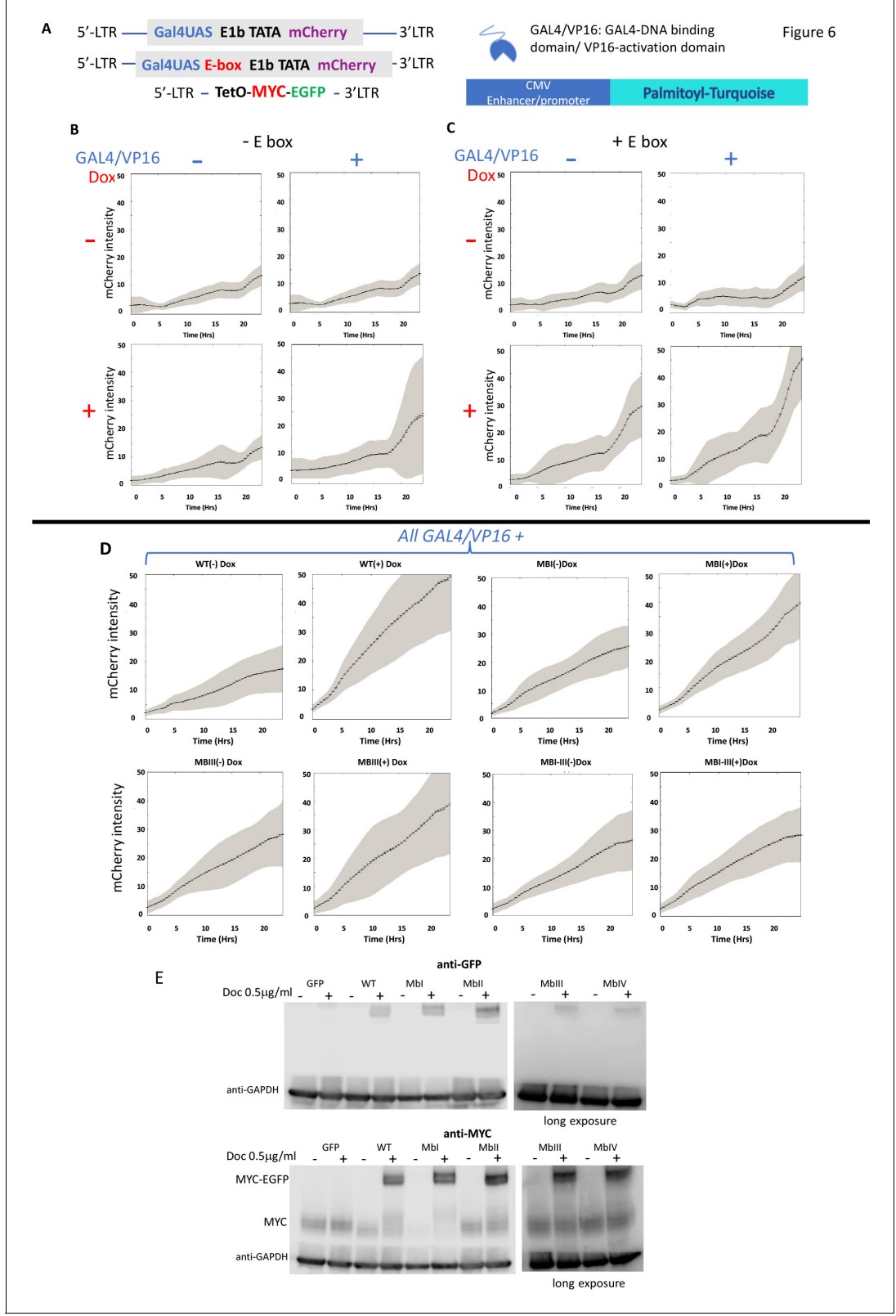

**Figure 6.** MYC amplifies the expression of chromosomally-integrated integrated reporters. (**A**) Lentiviruses encoding a fluorescent reporter driven by a TATA-box and a single Gal4UAS (top) or a single Gal4UAS and an E-box (middle) were inserted into cells bearing a doxycycline-inducible MYC-EGFP (bottom). Expression was monitored microscopically in single-cells. (**B**) Expression without an E-box requires both Gal4/VP16 and induced-MYC-EGFP. (**C**) E-boxes supplement but are not essential for MYC amplification. (**D**). MYC-amplification is resilient to single MB mutation. Lentivirus-expressed WT

*Figure 6 continued on next page*

*Figure 6 continued*

or MB-mutant MYC-EGFP induced by doxycycline were and transfected GAL4/VP16 were used to activate mCherry driven by a GAL4UAS and a single E-box. Error bars along the graphed lines indicate SEM and the shaded areas indicate the SD. (**E**) Western blot of uninduced and doxycycline-induced WT and MB-mutant MYC-EGFP relative to the endogenous MYC. Representative experiments are shown. For all experiments n ≥ 3.

individual MYC boxes. Attenuating wild-type MYC activity required MB mutation in combination (*Figure 6D*). Note that the lentivirus-MYC-EGFPs were expressed at levels above endogenous MYC but below those attainable with transient transfection (*Figure 1B* vs. *Figure 6E*, and not shown; note that expression of MYC-EGFP and mutants thereof was equivalent when assessed with anti-MYC or with anti-GFP). These amounts of MYC-EGFP were below those associated with the sharp upward inflection of reporter activity driven by transfected MBIII- or MBIV-mutants (*Figure 1B* and *Figure 3I*). (Also note that all lentivirus MYC and its mutants repressed the levels of endogenous MYC.)

## Wild-type MYC-EGFP binding at reporter promoters, parallels transcription amplification

The gain on MYC-amplified transcription increases as pre-amplified reporter expression goes up (see also *Figure 1—figure supplement 3A*). To test whether this augmented amplification is accompanied by enhanced MYC binding, ChIP was performed on the cells carrying chromosomally integrated lentivirus reporters driven by synthetic promoters, *GAL4UAS* and *GAL4UAS/E*. Cells were co-transfected with plasmids directing the expression of Gal4-VP16 and palmitoyl-mTurquoise, and then treated or not with doxycycline to induce MYC-EGFP. Transfected versus untransfected cells were sorted by flow cytometry gating upon mTurquoise. MYC-EGFP binding at promoters of sorted populations was monitored by ChIP using anti-GFP. Without Gal4/VP16, and without doxycycline treatment, uninduced MYC-EGFP binding at the integrated E-box-less promoter did not rise above the background (*Figure 7A*, lanes 1 vs. 2). Inclusion of an E-box licensed low levels of MYC-EGFP binding (*Figure 7A*, lanes 5 vs. 6). Also, without GAL4/VP16, induction with Dox modestly augmented MYC-EGFP recruitment to the reporter-promoter both in the absence and presence of an E-box (*Figure 7A*, lanes 3 vs. 4, 7 vs. 8).

Expressing GAL4/VP16 without MYC-EGFP induction permitted low levels of basal MYC-EGFP to bind at both non-E box and E-box promoters (*Figure 7A* lanes 9 vs. 10, 13 vs. 14). Inducing MYC-EGFP while expressing Gal4/VP16 dramatically enhanced the amount of MYC-EGFP bound at promoters (*Figure 7A*, lanes 11 vs. 12, 15 vs. 16) especially the E-box-bearing reporter (*Figure 7A*, lanes 15 vs. 16).

## MYC-Box mutations uncouple DNA binding from transcription activation

As noted above (*Figure 3J*), MYC-levels seemed to be inversely related to the apparent amplifier-gain. To see if mutant-MB MYC-EGFP binding at an endogenous promoter paralleled promoter output, expression and MYC-binding at the native *EZH2* promoter were monitored. *EZH2* binds and is induced by MYC (*Neri et al., 2012*; *Nie et al., 2012*). *MYC-EGFP* and mutant MB-*MYC-EGFPs* were induced with doxycycline and EZH2 expression was measured with RT-qPCR. As expected, *EZH2* expression was increased by wild-type MYC and mutant-MBIII compared with integrated empty vector, whereas expression was not increased by mutants -MBI, -MBII or -MBIV (*Figure 7B*). (Again, it should be noted that the levels of lentivirus-expressed mutant-MBIII and -MBIV did not reach the high transient expression levels that seemed to drive over-amplification in *Figure 3* and *5*). ChIP-PCR confirmed increased binding of wild-type MYC-EGFP at lentivirus mCherry promoter and the *EZH2* promoter following induction with doxycycline (*Figure 7C*); MYC-binding was not increased at the silent *BEX* gene (*Figure 7C*). Mutant-MBII MYC bound to EZH2 the same or better than wild-type despite being impaired for transcriptional activity (*Figure 7D*); augmented binding with diminished activity upon MBII mutation has been reported previously (*Farrell et al., 2013*). Surprisingly, notwithstanding its clear ability to increase promoter output, binding by mutant-MBIII MYC-EGFP not only did not increase but may have been diminished upon induction (*Figure 7D*), thus inviting consideration that it is productively removed from target genes as part of the transcription-cycle.

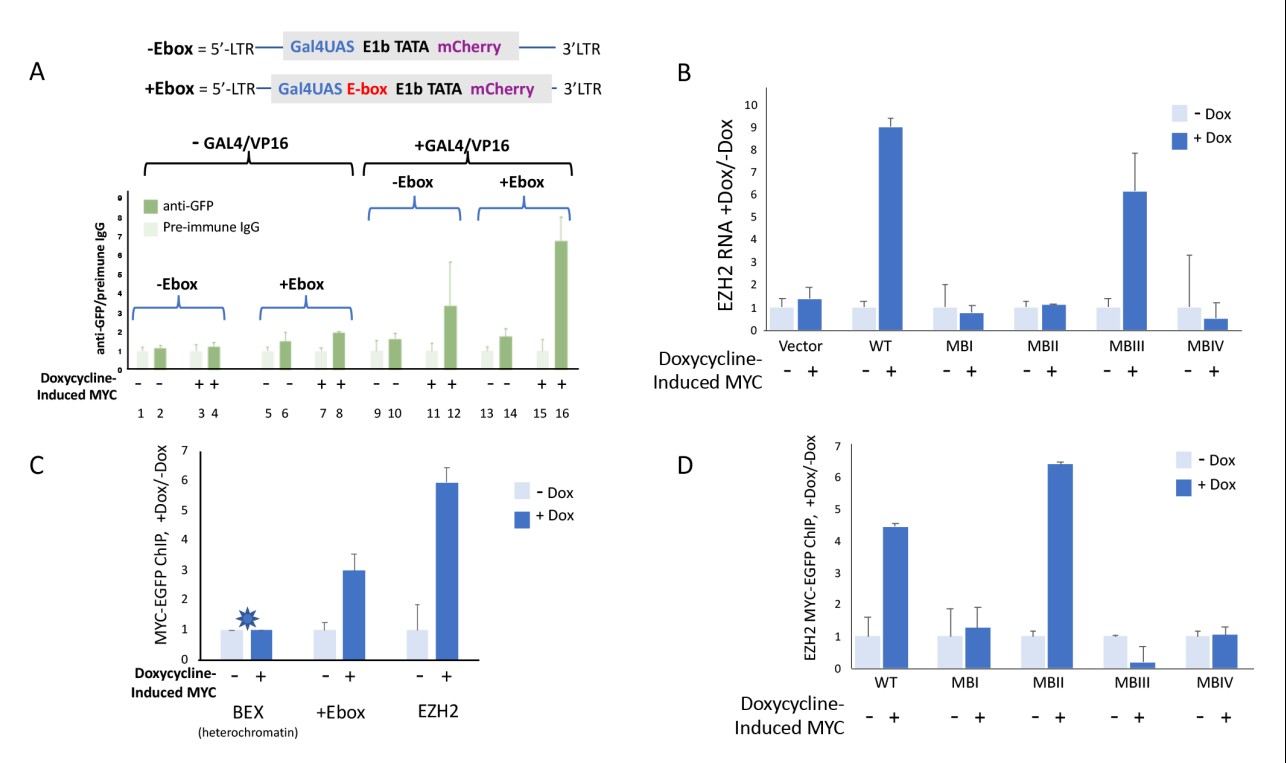

**Figure 7.** MYC-EGFP binding at chromosomally integrated-reporters parallels activity and emulates native promoters. Top: diagram of integrated-lentivirus CFP reporters. (A) Left-ChIP-PCR to assay MYC-EGFP binding in the absence of activator of uninduced (lanes 2 and 6) versus doxycycline-induced (lanes 4 and 8) MYC-EGFP, at non-E-box (lanes 2 and 4) versus E-box-bearing (lanes 6 and 8) lentivirus-integrated reporter promoters. Right-ChIP-PCR to assay MYC-EGFP binding in the presence of GAL4/VP16 of uninduced (lanes 10 and 14) versus doxycycline-induced (lanes 12 and 16) MYC-EGFP at non-E-box (lanes 10 and 12) versus E-box bearing (lanes 14 and 16) lentivirus integrated reporter-promoters. (B) *EZH2* RNA levels from cells expressing non-induced or doxycycline induced MYC-EGFP(WT) or MYC-EGFP-mutants MBI, MBII, MBIII or MBIV and assayed by RT-qPCR using Luna Universal One-Step RT-qPCR Kit (New England Biolabs). (C) Binding of uninduced and induced MYC-EGFP to endogenous promoters. The asterisk (*) indicates that binding to *BEX* in heterochromatin was so low as to be virtually unexpressed as previously reported (*Nie et al., 2012*). (D). Uninduced or doxycycline-induced MYC-EGFP(WT) or MYC-EGFP-mutants (MBI, MBII, MBIII, or MBIV) binding to the native EZH2 promoter assayed by ChIP-PCR. The mean and SD for representative experiments performed in triplicate are shown, n ≥ 2.

These data suggest that in vivo, the kinetics of MYC binding, action and degradation at promoters are complex.

## Defining MYC kinetics in transcription

The biochemical mechanism of MYC amplification of GR transactivation can be probed using a validated kinetic model of transcription (*Blackford et al., 2014*; *Blackford et al., 2012*; *Chow et al., 2015a*; *Chow et al., 2011*; *Chow et al., 2015b*; *Chow and Simons, 2018*; *Dougherty et al., 2012*; *Luo et al., 2013*; *Ong et al., 2010*; *Pradhan et al., 2016*; *Zhang et al., 2013*; *Zhu et al., 2014*) in the GREtkLUC system under the influence of the MYC inhibitor F4 and the well-studied transcriptional cofactor sSMRT, which has been categorized as a competitive decelerator (*Blackford et al., 2012*; *Dougherty et al., 2012*; *Zhang et al., 2013*). The model, which is described in more detail in the Supplementary Material, considers transcription as a sequence of complex-forming reactions and delivers predictions about the biochemical kinetic mechanisms of transcription factors in the language of enzyme kinetics. It predicts the site(s) of action of factors with respect to each other and relative to a special rate limiting reaction termed the 'concentration limiting step' (CLS). The hallmark of steroid-induced transcription, irrespective of the presence of MYC, is that the dose-response curve of the steroid agonist has a Hill coefficient of one as in a Michaelis-Menten reaction. This is unexpected for a sequence of reactions and requires that: 1) factors participating in reactions before the CLS are limited with respect to their binding affinity and thus downstream complexes must have

lower concentrations or lifetimes with respect to upstream complexes; 2) the CLS is the last of the reactions with limited factor; and 3) factors for steps after the CLS are more abundant than the limiting factor at the CLS. Finally, all reactions after the CLS can lead directly to the gene product. The dose response is fully characterized by the saturated maximal activity, $A_{max}$, and the steroid concentration of half-maximal activity, $EC_{50}$; explicit formulas for how these quantities change with respect to concentrations of the factors can be derived and compared to experiments (*Blackford et al., 2014*; *Blackford et al., 2012*; *Chow et al., 2015a*; *Chow et al., 2011*; *Chow et al., 2015b*; *Chow and Simons, 2018*; *Dougherty et al., 2012*; *Luo et al., 2013*; *Ong et al., 2010*; *Pradhan et al., 2016*; *Zhang et al., 2013*; *Zhu et al., 2014*).

With increasing MYC, $A_{max}$ (and $A_{max}/EC_{50}$) increase supra-linearly with varying concentrations of competing F4 (*Figure 8A and B*) or sSMRT (*Figure 8D and E*) whereas $1/EC_{50}$ remains relatively invariant (*Figure 8C and F*). Also, $A_{max}/EC_{50}$ is nonzero for zero MYC protein. As shown in the Appendix 1, which is based on mathematical derivations in references (*Blackford et al., 2014*; *Blackford et al., 2012*; *Chow et al., 2015a*; *Chow et al., 2011*; *Chow et al., 2015b*; *Chow and Simons, 2018*; *Dougherty et al., 2012*; *Luo et al., 2013*; *Ong et al., 2010*; *Pradhan et al., 2016*; *Zhang et al., 2013*; *Zhu et al., 2014*), there are at least two possible conclusions for the kinetic action of MYC. The first is that MYC is an accelerator acting after the CLS at two (or more) locations and MYC has relatively low binding affinity. The second is that MYC is an accelerator or a *facilitator* of a second accelerator acting at the CLS. In previous experiments, the CLS reaction has consistently been the binding of activated receptor-steroid complex to transfected GREtkLuc reporter, which is logical as the reporter is the most limited essential accelerator for transcription. Thus, MYC could act as an accelerator after the binding of GR to GRE or be an accelerator/facilitator of GRE through another actor (*Figure 8G*). The corepressor F4 reduces $A_{max}$ and $A_{max}/EC_{50}$, even in the absence of MYC. A plot of $EC_{50}/A_{max}$ vs. F4 is best fit by a quadratic plot (*Figure 8—figure supplement 1*). This indicates that F4 is acting at two sites, either as a competitive inhibitor of an accelerator (i.e. competitive decelerator) before or at the CLS or as an inhibitor of an activator of GRE. The second option is consistent with F4's action as a known inhibitor of MYC and MYCN (*Müller et al., 2014*). Thus, the second option predicts that MYCN is also an activator of GRE. sSMRT also acts as a competitive decelerator but the data were insufficiently precise to make a firm prediction of the location of action.

The results of the MYC-Box mutations that do not alter binding yet change amplification by MYC support the second mode of action, which is that MYC is a facilitator of an accelerator. This action is also largely independent of MYC's binding affinity to its promoter. This interpretation predicts that MYC's acts to facilitate the binding of other transcription factors.

## Discussion

Abundant evidence has indicted supraphysiological MYC levels in cancer. Yet the site(s) of MYC action and its physiological and pathological mechanisms remain controversial. On the one hand, MYC has been considered a sequence-specific transcription factor turning on particular genes via E-boxes and controlling a large set of genes comprising a dense overlapping regulon (*Alon, 2007*). On the other hand, MYC has been proposed to amplify the output of all expressed genes, with amplifier gain increasing along with output until saturation with E-boxes slightly biasing toward increased expression. In this scheme MYC upregulates all regulons. '-Omics' studies of RNA expression and transcription, and of genome binding by MYC, RNAPII and other factors have not resolved this conflict as the computational models employed for the analysis and filtering of data, and the selection of adjustable parameters, dictate what is observable. For example, though the DESeq2 algorithm (*Love et al., 2014*) is not suitable to gauge global changes in expression (only relative changes between a subset of genes) https://support.bioconductor.org/p/88413/, it has nevertheless been used in studies (*Muhar et al., 2018*; *Tesi et al., 2019*) that purport to refute transcription amplification by MYC. Therefore, we sought an experimental test of global amplification.

Here we have interrogated the MYC-amplifier hypothesis using synthetic biology experiments to study the influence of supraphysiological concentrations of MYC at minimal promoters, with or without E-boxes and with or without a separately recruited transactivator to provide an amplifiable input signal.

A number of observations support the notion that MYC is a general amplifier of transcription:

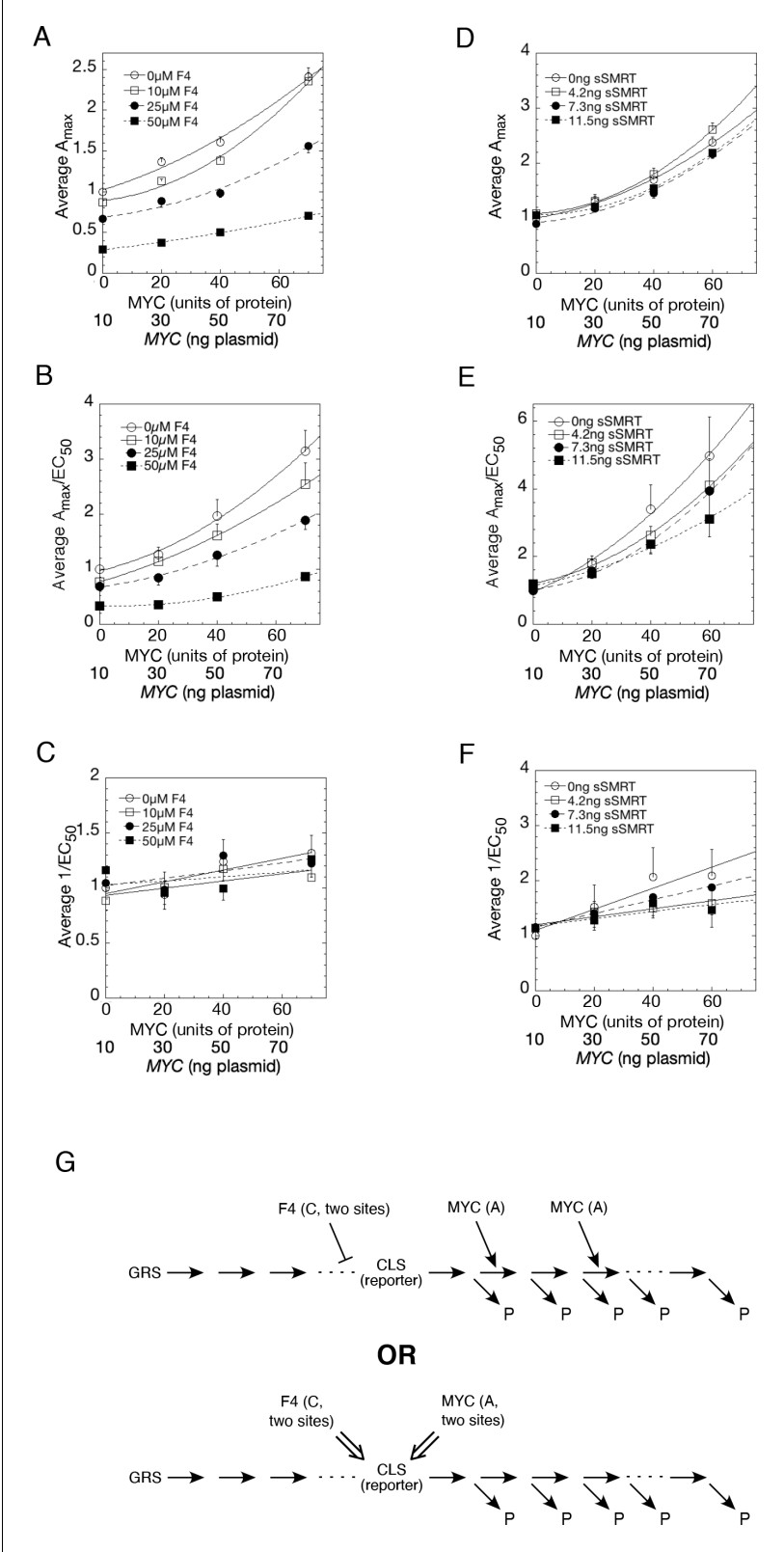

**Figure 8.** Competition assays reveal site of MYC action in the transcription-cycle in relation to other factors. (A – C) Competition of MYC vs 10058-F4. Double reciprocal plots (averaged) showing $A_{max}$ (A), $A_{max}/EC_{50}$ (B), and $1/EC_{50}$ (C) of MYC with varying inhibitor 10058-F4 from competition assays (as described in Materials and methods) in U2OS cells with *MYC* and co-transfected F4, GR, and GREtkLUC. Data from each experiment (n = 4) were

*Figure 8 continued on next page*

*Figure 8 continued*

normalized to the value for the lowest amounts of *MYC* and 10058-F4 and the averages (± SEM) were plotted. (**D–F**) Competition of MYC vs sSMRT. Double reciprocal (averaged) plots of $A_{max}$ (**D**), $A_{max}/EC_{50}$ (**E**), and $1/EC_{50}$ (**F**) with varying *sSMRT* were made for the competition assays, as in A-C, of *MYC* and decelerator *sSMRT* along with co-transfected GR and GREtkLUC. Data from each experiment (n = 5) were normalized to the value for the lowest amount of *MYC* and *sSMRT* and the averages (± SEM) were plotted. It should be noted that these plots require that the amounts transfected plasmid shown on the X-axis must correspond to the total amount of protein (endogenous plus exogenous) or added chemical. As shown in **Figures 1B**, 10 ng of MYC cDNA corresponds to zero total MYC protein. Therefore, the X-axis origin corresponds to 0 MYC protein, which equals 10 ng MYC cDNA. The data of the competition experiments of MYC vs 10058-F4 of Fig. 8A-C were expressed as $EC_{50}/A_{max}$, averaged, and plotted as averaged ± SEM (n = 4). The $R^2$ for the curve fits are all > 0.99. (**G**) Cartoon of sites and modes of action of MYC and 10058-F4 (F4). Starting with the Dex-bound GR (GRS), the induction of LUC protein (P) proceeds through numerous, undefined steps (arrows and ". . .") both before and after the CLS. After the CLS, each step can either proceed to protein or involve another regulatory cofactor before ending in the production of protein. The top panel depicts F4 acting as a competitive decelerator at two steps before or at the CLS while MYC acts as an accelerator at two steps after the CLS. The bottom panel is for MYC acting at two sites as an activator of GRE as it enters the CLS with F4 inhibiting MYC at two sites.

The online version of this article includes the following figure supplement(s) for figure 8:

**Figure supplement 1.** Competition assays reveal site of MYC action in the transcription cycle in relation to other factors.

1. Basal reporter expression is relatively insensitive to MYC. To increase output, a sufficient signal must first prime the promoter for MYC.
2. MYC can amplify a broad range of transactivators, for example steroid receptors and GAL4/VP16 that transduce a wide range of intra- and extracellular signals.
3. MYC drives reporters to much higher levels than are possible even with saturating levels of transactivator alone. So, supplying MYC is not equivalent to boosting the dose of transactivator. Beyond saturation with transactivator, reporter expression declines due to squelching. MYC suppresses squelching, dramatically elevating reporter output.
4. MYC amplification is auto-limited by MBIII, as evidenced by the much higher gain of MBIII mutants, especially at high levels of MYC. This peculiar property is not only rationalized but required by global amplification. If MYC increased gene transcription according to the level of ongoing promoter activity, then MYC-amplified transcription itself would become the substrate for further MYC-amplification on successive rounds of transcription. Unless checked, such a positive feedback loop delivers an all-or-none output that poorly matches the intensity of the transduced input signal. Suppression of positive feedback allows the output to follow the input across a broader dynamic range.
5. MYC acts kinetically as an accelerator acting at more than one location after the CLS or as a facilitator of an accelerator.

In this study we used well-characterized minimal promoters to simplify reporter expression and distill the fundamental role of MYC. In genomic studies, MYC amplifier activity has been inferred across a large spectrum of genes sorted both according to their expression and their levels of bound MYC. Such metagene analysis overcomes gene-by-gene variation but muddles the relationship between MYC binding and promoter output. Because transcription amplification is dependent on input signals, the panel of *cis*-elements arrayed along a gene, and the availability and activities of their corresponding transfactors, make the MYC transfer function at each promoter context dependent. MYC acts on top of these combinations of other regulatory molecules. That some promoter-recruited factors, such as MIZ1 (**Wiese et al., 2013**), may act locally to blunt MYC amplification reflects the complexity of metazoan gene regulation, but does not detract from or redefine the basic functional role of MYC.

The activators studied here, (GR, PR and GAL4/VP16), have not been reported to share coactivators nor to funnel directly through a shared regulatory complex onto a single discrete point in the transcription cycle. So MYC amplification of transactivation by GR or GAL4/VP16 is unlikely to reflect synergy driven through a common partner but is more likely to be accounted for by kinetic synergy (**Herschlag and Johnson, 1993**). More specifically, efficient up-regulation of a multi-step process such as the transcription-cycle, that transitions through several intermediate stages occurring on

roughly the same time-scale, requires the acceleration of each of these stages (*Chung and Levens, 2005*; *Herschlag and Johnson, 1993*). An agent that accelerates a single reaction in a chain of kinetically equivalent steps has limited capacity to increase the output of the end-product yet synergizes effectively with factors that quicken the other steps (*Chung and Levens, 2005*). Note that in the case of transcription, there is no requirement that these various accelerators associate physically nor even contemporaneously co-occupy a promoter. MYC interacts with such a broad spectrum of gene regulatory components that it has the potential to modify virtually any step in the transcription cycle (*Agrawal et al., 2010*; *Kalkat et al., 2018*) to minimize kinetic bottlenecks and increase promoter output.

The mutational analysis of MBs I-IV is consistent with MYC-driven kinetic synergy operating throughout the transcription cycle at minimal promoters. No single MB mutation eliminated transactivation by MYC but, to different degrees, each individual mutation modified reporter expression. That reporter expression was refractory to single MB mutations must reflect either that the inactivation of the targeted MBs is incomplete or that transcription amplification is sustained by the non-targeted MBs and/or other portions of MYC. The crippling of amplification by MB double and triple-mutants argues that multiple segments of MYC cooperate to enforce higher output. Much biochemical evidence indicates that each of the MBs associates with or is in close proximity to a characteristic set of proteins. These various protein sets function at different stages of chromatin opening and the transcription cycle. The uncoupling of binding and transcription activation by mutant MBs suggests that MYC might act sequentially at several steps, cycling on and off the promoter (by dissociation, release or degradation) during a single round of transcription. The MBIII mutation that leads to hyperactivity seems also to strip MYC from the *EZH2* promoter. That co-mutation of MBI blunts this hyperactivation while stabilizing the protein argues for MYC-degradation as a step in MYC action, especially in view of MBI's proven role in MYC proteolysis (*Farrell and Sears, 2014*). Degradation or energy-dependent modification of components such as MYC may thermodynamically drive the transcription-cycle forward and improve regulatory precision (*Hopfield, 1974*; *Hopfield, 1980*). MBII mutation in contrast impairs reporter output while increasing both MYC protein levels and *EZH2* promoter binding, again consistent with activation-driven MYC-degradation (*Farrell et al., 2013*; *Farrell and Sears, 2014*; *Kim et al., 2003*; *Molinari et al., 1999*; *Salghetti et al., 2000*).

The minimal promoters employed in our studies are devoid of characterized positive-acting *cis*-elements and are insulated from enhancers, hence they are poorly equipped to recruit and deploy the components that expedite transit of the transcription machinery through the transcription cycle. Consequently, these promoters depend upon MYC to supply the complexes that sustain higher expression. In contrast, at high output promoters studded with strong positive *cis*-elements, multiple transfactors may be recruited that modify and complicate the kinetic landscape upon which MYC acts. A major advantage of the minimal promoters, primed by specific transactivators used here, is that it should greatly reduce the possible sites at which MYC exerts its kinetic activity(s). In fact, if a factor acts at multiple steps in a specific reaction pathway, our model will not give a single mechanistic interpretation from the competition assay (*Chow and Simons, 2018*). The ability to interpret the observed data in terms of a single, internally consistent kinetic mechanistic description is very strong evidence that the model is applicable and the major action of the factor under the experimental conditions being examined proceeds *via* that mechanism. Therefore, while MYC is certainly capable of acting at a variety of sites *via* different kinetic mechanisms, we can safely conclude that, under the conditions of *Figure 8*, MYC acts predominantly as a facilitator of one or more accelerators acting at or after the CLS. This interpretation is compatible with both the widely held view that MYC acts late in the transcription-cycle to affect pause-release (*Baluapuri et al., 2019*; *Jaenicke et al., 2016*; *Nie et al., 2012*; *Rahl et al., 2010*) and with prior and recent reports that MYC acts earlier during transcription initiation (*Barrett et al., 2005*; *Tesi et al., 2019*).

The conclusion that an inhibitor of MYC action (i.e., 10058-F4) acts as a competitive decelerator at two steps before the CLS, and before MYC, is not unreasonable in the absence of conflicting data. The actions of 10058-F4 can still be specific for MYC if, for example, 10058-F4 prevents the pre-CLS entry of MYC into the reaction pathway or if it negatively affects some factor that is required by MYC at its further downstream site of action. The exposition of the exact biochemical reactions that are associated with the formal kinetic steps regulated by MYC promises to expose therapeutic vulnerabilities to intercept the pathology of this oncogene.

# Materials and methods

### Key resources table

| Reagent type (species) or resource | Designation | Source or reference | Identifiers | Additional information |
|---|---|---|---|---|
| Gene (*Homo sapiens*) | c-Myc | GeneBank | | |
| strain, strain background (*Escherichia coli*) | DH5a | NEB | Catalog #C2987I | High efficiency competent cells |
| strain, strain background (*Escherichia coli*) | XL-10-Gold | Agilent | Catalog #200314 | Ultracompetent Cells |
| genetic reagent (include species here) | | | | |
| cell line (*Homo sapiens*) | U-2 OS | ATCC | ATCC HTB-96 | Lack glucorticoid response |
| cell line (*Homo sapiens*) | Lentivirus-expressed MYC, mCherry-reporter double-stable cell lines in U-2OS | This paper | bone osteosarcoma | Cells maintained in D.Levens lab |
| cell line (*Rattus norvegicus* (Rat)) | Myc-ER HO15.19 | RRID:CVCL_0311 *O'Connell et al., 2003* | | |
| transfected construct | Palmitoyl-turquoise2 | gift from Dorus Gadella; | Addgene plasmid # 36209 | http://n2t.net/addgene *Goedhart et al., 2012* |
| transfected construct | Pd4EGFP-c-Myc and Myc mutants | This Paper | | |
| transfected construct | PTripZ-EGFP-cMyc and mutant | This paper | | |
| transfected construct | PLVX-PKG-GAL4USA-(Ebox)-E1bTATA-mcherry-hygro (Plasmid) | This Paper | | |
| biological sample (include species here) | | | | |
| antibody | anti-cMyc (rabbit monoclonal) | Abcam | Catalog #ab-32072 | 1:5000 |
| antibody | anti-GFP rabbit polyclonal | Abcam | Catalog #ab-32146 | 1:1000 |
| antibody | anti-GAPDH | Abcam | Catalog #ab-9485 | 1:2500 |
| recombinant DNA reagent | Pd4EGFP | This paper | | |
| recombinant DNA reagent | PTripZ-EGFP | This paper | | |
| recombinant DNA reagent | pLVX-mCherry-N1 | Takara | Catalog #632562 | |
| sequence-based reagent | PKG-F | This Paper | CR primer | GTGAGCGGCCGCGACTCTGAGTAATTCTACCGGGTAGG |
| sequence-based reagent | PKG-R | This Paper | PCR primer | GTGAGCGGCCGCGACTCTGAGTAATTCTACCGGGTAGG |

*Continued on next page*

*Continued*

| Reagent type (species) or resource | Designation | Source or reference | Identifiers | Additional information |
|---|---|---|---|---|
| sequence-based reagent | Hygro1-F | This Paper | PCR primer | GCCCAAGCTTACCATGAAAAA GCCTGAACTCACC |
| sequence-based reagent | Hygro1-R | This Paper | PCR primer | TGTTGGAGCCGAAATCCGCGTGCA |
| sequence-based reagent | Hygro2-F | This Paper | PCR primer | GGATTTCGGCTCCAACAATGTC CTGACGGACAATG |
| sequence-based reagent | Hygro2-R | This Paper | PCR primer | GGATTTCGGCTCCAACAATGTC CTGACGGACAATG |
| peptide, recombinant protein | | | | |
| commercial assay or kit | Direct-zol RNA miniprep kit | Zymo Research | Catalog #R2051 | |
| commercial assay or kit | Luna Cell Ready One-Step RT-qPCR Kit | NEB | Catalog #E3031, #E3005, #E3006 | |
| commercial assay or kit | Lipofectamine 2000, 3000 transfection kit | Thermo-Fish (Invitrogen) | Catalog #L2000, L3000-008 | |
| commercial assay or kit | Lenti-X Accelerator kit | Takara | Catalog #631257 | |
| commercial assay or kit | Lenti-X Packaging Single Shot kit | Takara | Catalog #631275 | |
| commercial assay or kit | the ChIP-IT High Sensitivity kit | Active Motif | Catalog #53040 | |
| chemical compound, drug | Dexamethasone (Dex) | Sigma | | |
| chemical compound, drug | Promegestone | PerkinElmer Life Sciences | R5020 | |
| chemical compound, drug | 10058-F | Sigma | Catalog #F3680 | |
| chemical compound, drug | 4-OH tamoxifen | Sigma-Aldrich | Catalog #94873 | |
| chemical compound, drug | All the restriction enzymes and DNA polymerase | NEB | | |
| software, algorithm | KaleidaGraph | Synergy Software, Reading, PA | | |
| software, algorithm | NIS-Element | Nikon | | |
| software, algorithm | MATLAB | Mathworks | | |

*Continued on next page*

*Continued*

| Reagent type (species) or resource | Designation | Source or reference | Identifiers | Additional information |
|---|---|---|---|---|
| software, algorithm | LightCycler 480 Probes master | Roche-applied Science | ref number 04707494001 | |
| other | | | | |

## Chemicals

Dexamethasone (Dex) was obtained from Sigma (St. Louis, MO) and promegestone (R5020) was from PerkinElmer Life Sciences (Boston, MA). Restriction enzymes and DNA polymerase were purchased from New England Biolabs (Beverly, MA). MYC inhibitor, F4 (5-[(4-ethylphenyl)methylene]−2-thioxo-4-thiazolidinone; 10058-F4)), was from Sigma, St. Louis, MO (Cat. No. F3680). 4-OH tamoxifen was from Sigma-Aldrich, catalog No.94873.

## Plasmids

Renilla-TS reporter, GREtkLUC, and rat GR (pSG5/GR) have been previously described (*Wang et al., 2004*). The FR-LUC reporter, in this paper referred to as 5*xGAL4UAS-TATALuc,* is from Stratagene (La Jolla, CA). E-box GRE tkLUC was constructed by amplifying the fragment in GREtkLUC with forward primer 5'-GCACcatatgCGGTGTGAAATACCGC and reverse primer 5'-ACCgtcgacCACGTGCTGCAGGCATGCAAGCTTG using Nde I and Sal I respectively to insert the E-box CACGTG. The resulting E-box GREtkLUC was sequenced and confirmed by using sequencing primer 5'-TTCGAGGCCACACGCGTCAC-3'. MYC mutants were generated from p-c-Myc-pd4-EGFP-N1 (*Nie et al., 2012*) by replacing the amino acids IWKKE (aa49-53) in MBI, DCMW (aa132-135) in MB II, EIDVV (aa261-265) in MB III, and QHNY (aa306-309) in MB IV with the same number of alanines, respectively. All double-, triple- MB box mutants were constructed by mutating the single-, double-, triple- mutants, in different combinations. The MYC box mutated plasmids were generated by replacing the wild type *MYC* sequences with G-blocks (IDT) containing MB mutants using Gibson Assembly Cloning Kit (NEB, E5510). The sequences of G-blocks are shown in *Supplementary file 1*.

An HIV-1 based, Lentiviral expression vector PLVX-mcherry-N1 was used to generate integrating reporter constructs. To temper reporter expression, the ClaI-XhoI restriction fragment of human cytomegalovirus immediate early promoter Pcmv IE was replaced by a minimal promoter containing a Lex binding site, Gal4 binding site(G), and an E1b TATA box with or without an E-box. The sequences of the fragments are listed below. Each pair of fragments were annealed at 96℃ for 5 min with 1x NEB buffer #2 and slowly cooled to the room temperature before further subcloning into the vector yielding two reporter constructs yielding *PLVX-Gal4UAS-E1bTATA mCherry* and *PLVX-Gal4UAS-E-box-E1bTATAmCherry*.

1. ClaI-LexGalE1bTATA-XhoI-T: CGATAGGTACTGTATGTACATACAGTACTCGACTGCAGTCGGAGGACAGTACTCCGCTAGCCCGACTAGAGTCGAGTACTGTATATAATC;
2. ClaI-LexGalE1bTATA-XhoI-B: TCGAGATTATATACAGTACTCGACTCTAGTCGGGCTAGTCGGGCTAGCGGAGTACTGTCCTCCGACTGCAGTCGAGTACTGTAGTACATACAGTACCTAT;
3. ClaI-LexGalE-boxE1bTATA-XhoI-T: CGATAGGTACTGTATGTACATACAGTACTCGACTGCAGTCGGAGGACAGTACTCCGCTAGCCCGCACGTGAGAGTCGAGTACTGTATATAATC;
4. ClaI-LexGalE-boxE1bTATA-XhoI -B: TCGAGATTATATACAGTACTCGACTCTCACGTGCGGGCTAGCGGAGTACTGTCCTCCGACTGCAGTCGAGTACTGTATGTACATACAGTACCTAT;

## Cell lines

In order to make double stable cell lines, the puromycin-resistance genes *PLVX-Gal4UAS-E1bTATA mCherry* and *PLVX-Gal4UAS-E-box-E1bTATAmCherry* were replaced by the Hygromycin resistance gene using thermostable assembly. The reporter genes were digested with XbaI and TthIlli restriction enzymes, gel purified fragment then assembled with PCR products of PGK promoter (554 bp),

Hygro I (622 bp), and Hygro2 (459 bp). The resulting constructs were used to make double stable cell lines. The primers used for PCR are in below.

1. PKG-F5'-GTGAGCGGCCGCGACTCTGAGTAATTCTACCGGGTAGG-3',
2. PKG-R5'-TCATGGTAAGCTTGGGCTGCAGGTCGAAAGG-3',
3. Hygro1-F5'-GCCCAAGCTTACCATGAAAAA21GCCTGAACTCACC-3',
4. Hygro1-R5'-TGTTGGAGCCGAAATCCGCGTGCA-3',
5. Hygro2-F5'-GGATTTCGGCTCCAACAATGTCCTGACGGACAATG-3',
6. Hygro2-R5'-GGATTTCGGCTCCAACAATGTCCTGACGGACAATG-3'

## Cell culture, transfection and lentivirus infection

U2OS human osteosarcoma cells were from ATTC, #HTB-96. These cells were maintained without further verification or mycoplasma testing, but retained proper growth characteristics, morphology, and properties. Key for these studies, they lacked an intrinsic glucorticoid response. Monolayer cultures of U2OS cells were grown in high glucose (4.5 g/L) DMEM containing 10% FBS in a 37°C incubator supplied with 5% oxygen-air atmosphere. Cells are transfected for 18 hr using Lipofectamine (Life Technologies, Inc, Gaithersburg, MD). For each well of a 24-well plate, 100 ng of reporter (*FRLUC*, *GRE-E-box-tkLUC* or *GREtkLUC*) and 10 ng Renilla plus various combinations of other expression vectors (e.g., GR, cMyc, cMyc/EGFP) were used. Equal molar amounts of the respective empty expression vectors were included to keep the molar amount of each vector constant, with the total transfected DNA brought to 300 ng/well with pBluescript II SK (Stratagene).

The packaging of all *pTripZ c-Myc-GFP* and Myc mutants-GFP plasmids into lentiviral vector were performed as previously described (*Porter et al., 2017*). *The PLVX-LGET-Cherry-hygromycin* lentiviruses were produced following the manufacturer's instructions. The day before transfection, 293 cells were plated on 100 mm dishes at 60–80% confluency. Transfection was performed by adding 7 µg of DNA into the cells using Lenti-X Packaging Single Shot kit (Takara, cat. No.631275). Sixteen hrs. later, the medium was replaced with 14 ml of fresh full medium. Virus was harvested at 48 hr and 68 hr and filtered through 0.45 µm PVDF filters. The viruses were concentrated using the Lenti-X concentrator (Takara, cat. #631231). Concentrated viruses were then aliquoted and stored in −80°C.

For lentivirus reporter experiments U2OS cells were grown in DMSO containing 10% Tet-approved FBS (Takara, cat. No.631103), MEM Nonessential Amino Acids (Corning cellgro REF. 25–025 Cl) and 50 U/ml penicillin G, 50 µg/ml streptomycin sulfate (Gibco REF. 15070–063). All pTripZ containing stable U2OS cell-lines were established by transducing the viruses into U2OS cells followed manufacturer's protocols as previously described (*Porter et al., 2017*). The Lenti-X hygromycin containing viruses were transduced into pTripZ containing stable cell-lines with Lenti-X Accelerator kit (Takara, cat. No. 631257) according to the manufacturer's instructions. In general, the viruses were mixed with the Lenti-X accelerator beads for 20–30 min and then added to the pre-plated 60–80% confluent cells on the magnetic plate for about 5 min. After changing medium, the cells were incubated in 37°C, 5% CO2 incubator for 24 hr and then selected with either 0.8 mg/ml of puromycin or both 0.8 mg/ml of puromycin and 200 mg/ml hygromycin.

The lentivirus-expressing MYC (wild type or mutated) and mCherry-reporter double-stable cell lines were plated in glass bottom cultureware plates (MatTek P06G-1.5–14F) the day before transfection with 60–80% confluency. Transfection of 10–20 ng of Gal4-VP16, 35 ng of Palmitoyl-turquoise2 (gift from Dorus Gadella; Addgene plasmid # 36209; http://n2t.net/addgene:36209 ; RRID: Addgene_36209) (*Goedhart et al., 2012*) and pBluescript (SK) plasmid DNA for a total of 1 µg DNA, was carried out with the Lipofectamine 3000 transfection kit (Invitrogen catalog number L3000-008) according to the manufacturer's instructions) and subjected to image analysis and/or cell sorting.

## Measurement of RNA expression

For MYC-ER cell total RNA, rat Myc-ER cell line HO15.19 grown as previously (*O'Connell et al., 2003*) described in six well plates with DMEM and 10% of calf serum (CS). At about 70–80% of confluency, cells were starved with 0.1% of CS for about 5 days and then treated or not with 200 nm of 4-OH tamoxifen (Sigma-Aldrich, catalog No.94873) for 6, 10 and 13 hr respectively. Cells were then lysed and RNA extracted in 1x PBS, 2 mM EDTA, using the Direct-zol RNA miniprep kit (Zymo Research, Catalog No. R2051). RNA was measure using Nanodrop. *For measurement of human*

*EZH2 RNA* in pTripZ-EGFP, or pTripZ-lentivirus vector directed wild-type MYC-EGFP and mutant-MBI, II, III or IV-MYC-EGFP U2OS stable cell lines, RNA was prepared using Luna Cell Ready One-Step RT-qPCR Kit (NEB # E3031) following the manufacturer's instructions. ~ $0.2 \times 10^4$ cells/well of each were seeded on a 96 well plate and grown at 37°C, 5% $CO_2$ overnight. Cells were then treated or not with 0.5 mg/ml of doxycycline. After 8 hr., cell cultures were removed, rinsed with cold 1x PBS and aspirated. 25 µl Luna Cell Ready Lysis Buffer (2X), 5 µl of RNase-free DNase I (10x), 2 µl of Luna Cell ready RNA Protection Reagent (25x), 2 µl of Luna Cell Ready Protease (25x) and 6 µl of Nuclease-free water for total volume of 40 µl / well were added and incubated at 37°C for 10 min. The cell lysis reactions were terminated by adding 5 µl of Luna Cell Ready Stop Solution to each well and mixed well then incubated at 25°C for 5 min. 1.5 µl of the above cell lysis mix were used to program q-PCR reactions using Luna Universal One-Step RT-qPCR kit (NEB # E 3006). For a 20 µl reaction mixture, 10 µl of Luna Universal probe One Step Reaction Mix, 1 µl of Luna WarmStart RT Enzyme Mix, 0.8 µl each of the 10 mM primer, 0.4 µl of 10 mM probe, 1.5 µl of cell lysate and Nuclease-free Water were added. Triplicate reactions for each sample were performed. The PCR programs were set as 1 cycle of reverse transcription at 55°C for 10 min; 1 cycle of initial denaturation at 95°C for I min; 40 cycles of denaturation at 95°C for 10 s followed by extension at 60°C for 30 s for each cycle. The primers used for human EZH2 expression were: 5'-ATG GCA CCT GCA GAA GGA-3'; 5'-TTG GGA AGC CGT CCT CTT-3'. The probe used for the qPCR is Universal probe library # 79. *For measurement of Luc and Ren RNA* using qPCR, transfections were performed as above in 24-well plates and RNA was prepared with Luna cell ready one-step RT-qPCR kit (NEB # E3030). The cell lysis was as described for *EZH2* expression for MYC -EGFP and mutant MB MYC-EGFP except 120 ml of cell lysis mix was used and all other reagent were proportionally increased. Real time qPCR reactions were performed with Luna one-step RT-qPCR kit (NEB # E3005) according to the manufacturer's instructions. Primers were Firefly *luc* F 5'- GTG TTG GGC GCG TTA TTT AT-3', Firefly *luc* R 5'- CAT CGA CTG AAA TCC CTG GT-3'; *renilla* F 5'- GTG GTG GGC CAG ATG TAA AC −3', renilla R 5'- ACC AGA TTT GCC TGA TTT GC-3'. The SYBR scan mode was used on the real-time instrument. The instrument setting for reverse transcription is 55°C for 10 min. The initial denaturation step is at 95°C for 1 min followed by 40 cycles of denaturation at 95°C for 10 s. then 60°C for 30 s extension for each cycle. The melt curve was set for 60°C.

## Luciferase assay

U2OS cells were treated for 20–24 hr with/without steroid (100 nM in ethanol unless otherwise stated) in the culture media (final ethanol concentration was 0.1%) before being harvested in $1 \times$ Passive Lysis Buffer (150 µl/well in 24-well plate; Promega). Added 10058-F4 to a final concentration of 50 µM (from 100 mM in DMSO) was used as an inhibitor in some MYC amplification experiments. Cell lysates (50 µl per well) in a 96-well plate were used for luciferase activity assays using the Dual-Luciferase Assay System (Promega). Bioluminescence was read by a GloMax 96 Microplate Luminometer (Promega). The fold induction by steroid is defined as (the activity with steroid)/(the basal level activity seen in the absence of steroid). Each sample was run in triplicate and each experiment was performed at least two times. The luciferase induction curve with steroid was then analyzed using KaleidaGraph (Synergy Software, Reading, PA) for the best fit by Michaelis–Menten kinetics ($R^2$ usually $\geq 0.95$). All errors are ± S.E.M.

## Cell sorting

The cell growth and reactions conditions were the same as used for microscopic analysis. After 18 hr, the cells having GFP, Turquoise, and mCherry fluorescent proteins were washed twice with PBS and were strained using 50-micron pore size filters. Cell suspensions were analyzed and sorted using FACSAria-IIu cell sorter (BD Biosciences, USA) equipped with 355 nm, 407 nm, 488 nm, 532 nm, and 640 nm LASER lines using DIVA eight software (BD Biosciences, USA). Briefly, after excluding cellular debris, using a forward light scatter (FSC)/side scatter plot (SSC), doublet cells were removed on the basis of signal width of FSC and SSC parameters. These debris free single cells were further gated to identify fluorescence signals of GFP, Turquoise, and mCherry proteins. GFP fluorescence was excited by 488 nm LASER and emission was read using 515/20 nm bandpass filter sets on PMT (B515). Turquoise protein's fluorescence was excited by 407 nm LASER and emission was read using 450/50 nm bandpass filter sets on PMT (V450), whereas mCherry protein's fluorescence was excited

by 532 nM LASER and emission was read using 610/20 nm bandpass filter sets on PMT (G610). GFP-positive, but turquoise-negative (and hence GAL4/VP16-negative) vs. GFP-positive-turquoise-positive (and hence GAL4/VP16-positive) cells were sorted and stored at −80°C for ChIP-PCR analysis.

## Immunoblot

For transiently expressed MYC-EGFP, different amounts of *MYC-EGFP* plasmids were transfected into U2OS cells in 6-well plates for 48 hr using Lipofectamine (Life Technologies, Inc, Gaithersburg, MD). Cell lysates were collected and supernatants were obtained by centrifugation at 13,500 g for 20 min (4°C). Equal amounts of lysates protein were resolved into 3–8% Bis-acetate gels (Invitrogen) at 150 v and transferred onto nitrocellulose membranes (Bio-Rad) at 20 v at room temperature for 1 hr using a semi-dry apparatus (Bio-Rad). MYC or MYC-EGFP protein were detected using anti-MYC antibody from Epitomics (Rabbit mAb Cat.# 1472–1) and a secondary goat-anti rabbit IgG–HRP (Santa Cruz Biotechnology, Dallas, TX) antibody was used to amplify the signal. Western blots were visualized by ECL detection reagents as described by the manufacturer (Amersham Biosciences). The relative protein expression levels were compared by using a densitometer (Bio-Rad). *For immunoblot of cellular and stably expressed MYC-EGFP,* lentivirus expressed MYC-EGFP and MB-mutant MYC-EGFP, the U2OS cell lines with integrated lentivirus vectors expressing pTripZ-EGFP-, MYC-EGFP, or MYC-EGFPs with mutant MBI, MBII, MbIII or MbIV were grown in DMEM standard medium with 10% Tet-system approved FBS (Takara bio, Catalog No. 631106). After splitting into 6-well plates (~$3 \times 10^5$/well), cells were grown for another 22 hr with or without 0.6 mg/ml doxycycline for 8 hr. Cells were lysed with 200 ml of RIPA buffer with protease inhibitors at 4°C for 20 min. The cell-lysates were centrifuged at 4°C for 20 min and the supernatants were collected. Cell lysate containing 40 mg protein were used for western Blot analysis. Both anti-GFP (Abcam Catalog No. ab-32146) and anti-cMyc (Abcam Catalog No. ab-32072) were used for detection. Anti-GAPDH antibody (Abcam, ab9485) was used as a loading control.

## Time-lapse microscopy

Images were acquired on a Nikon TiE inverted fluorescence microscope equipped with an automatic focus correction system, a SOLA LED light source (Lumencor), and a Neo sCMOS camera (ANDOR Technology Ltd). The Nikon TiE fluorescent microscope was adapted for long-term time-lapse microscopy through the addition of an environmental chamber that maintains a constant environment of 37°C, 5% CO2, and 40% humidity. Images were acquired with a 20x plan apo objective (NA 0.75) every 30 min over a 24 hr period. The mCherry filter set contained filters of 550/28 nm for the excitation light, 585 nm for the dichroic beam splitter, and 605/52 nm for the emission light (Chroma). The GFP filter set contained filters of 485/25 nm for the excitation light, 495 nm for the dichroic beam splitter, and 525/36 nm for the emission light (Chroma). The Turquoise filter set contained filters of 434/21 nm for the excitation light, 450 nm for the dichroic beam splitter, and 480/40 nm for the emission light (Chroma). Images were analyzed using NIS-Elements software (Nikon) and MATLAB software (Mathworks), which is available upon request.

## Chip-pcr

The binding of MYC to the reporter and endogenous promoter regions was performed with the ChIP-IT High Sensitivity kit (Active Motif, Catalog number:53040) following the manufacturer's instructions with minor changes. Frozen cells were resuspended and fixed in the growth medium containing 1/10 vol of fixation buffer at room temperature for 15 min. with gently shaking and then 1/20 medium volume of stop solution was added. The chilled cells were lysed with 30 strokes through a 25½ G needle and then centrifuge at 1250 g for 3 min. Chromatin sonication was performed with Qsonica sonicator Q700 systems using microprobes. Sonication was performed at 30% amplitude, pulse 30 s on and 30 s off for a total ''on'' time of 10–20 min, depending on the cell number. Chromatin immunoprecipitation was performed with anti-GFP antibody (Abcam) or pre-Immune normal rabbit IgG (Santa Cruz Biotechnology, catalog number: sc-2017) as control. Quantitative PCR was performed with LightCycler480 probe system and the LightCycler 480 Probes master (Roche-applied Science, ref number: 04707494001). The primers for BEX and EZH2 are the same as previously described (*Nie et al., 2012*). The primers forPLVX-LGET-mCherry were: PLVX-LGET -F: 5′AAA TTCAAAATTTTCGGGTTTATTAC-3′, -R: 5′CGGGGCTAGCGGAGTACTGT-3′. The Universal library

probes used for the PCR reaction were PLVX-LGET-mcherry: #56; human EZH2; #82; human BEX: # 52. To compare WT MYC-EGFP and Myc -Box mutant MYC-EGFP binding to promoters, the procedures are the same as previously described (*Nie et al., 2012*) and above with some modifications. The wt and MBI, II, III, and IV mutated -MYC pTripZ EGFP stable U2OS cells each were grown in the 2 150 mm plates. At about 80–90% confluency, cells were treated with or without 0.5 mg/ml of doxycycline for 8 hr. The cells cross-linked with formaldehyde, sonicated and subjected to chromatin immunoprecipitation as before. The PCR-reaction conditions are antibody, primers and the reagents are the same as above.

## Two-factor competition assays

Detailed descriptions of the two-factor competition assay for steroid-induced gene transcription, and the resulting predictions of the mechanism and site of action of the factors being assayed, can be found elsewhere (*Blackford et al., 2014*; *Blackford et al., 2012*; *Chow and Simons, 2018*; *Dougherty et al., 2012*; *Ong et al., 2010*; *Pradhan et al., 2016*; *Zhu et al., 2014*) . Briefly, the experiments consist of measuring the gene induction dose-response curve as a function of Dex for various combinations of two added factors. From each factor combination the maximal activity $A_{max}$ and the Dex dose of half-maximum $EC_{50}$, are estimated from the dose-response curve. The basic protocol for gene induction (*Zhu et al., 2014*) was followed, where each assay consists of 16 combinations (4 concentrations of both factor 1 and factor 2), each with four concentrations of Dex, all in triplicate, for a total of 192 wells. Triplicate samples of cells were seeded into 24-well plates at 20,000 cells per well and transiently transfected the following day with luciferase reporter (GREtkLUC) and DNA plasmids for rat GR plus the factors being examined by using 0.7 µl Lipofectamine (Invitrogen) or Fugene 6 (Roche) per well according to the manufacturer's instructions. The total transfected DNA was adjusted to 300 ng/well of a 24-well plate with pBluescript II SK+ (Stratagene; La Jolla, CA). The molar amount of plasmids expressing different protein constructs was kept constant with added empty plasmid or plasmid expressing human serum albumin (*Wang et al., 2004*). Renilla-TS (10 ng/well of a 24-well plate) was included as an internal control. After transfection (32 hr), cells were treated with medium containing the appropriate chemicals and/or hormone dilutions. The cells were lysed 20 hr later and assayed for reporter gene activity using dual luciferase assay reagents according to the manufacturer's instructions (Promega, Madison, WI). Luciferase activity was measured by GloMax 96 Microplate Luminometer (Promega). The data were expressed as a percentage of the maximal luciferase response with Dex above background before being plotted ± standard error of the mean, unless otherwise noted. All plots of the data assume a linear increase in factor plotted on the x-axis. When Western blots revealed a non-linear relationship between the optical density of scanned protein band and the amount of transfected plasmid at constant levels either of total cellular protein, or of β-actin, the linear equivalent of expressed plasmid must be determined (*Dougherty et al., 2012*). In contrast to previous analyses, but consistent with the presentation of data in the rest of this paper, the values of $A_{max}$ are not corrected for Renilla expression because Renilla expression is itself affected by MYC. An unbiased estimate of the intersection point of a set of linear regression fits to graphs such as $A_{max}/EC_{50}$ vs. factor, is determined from what is called an 'a vs. b plot'.

## Acknowledgements

This work was supported by the Intramural Research Programs of the National Cancer Institute (Center for Cancer Research) and the National Institute of Diabetes and Digestive and Kidney Diseases. We thank the NHLBI Flow Cytometry Core and for DNA sequencing the CCR Genomics Core. Author Contributions: ZN, CG and designed and conducted experiments and analyzed data, SKD designed and conducted experiments, CCC analyzed data and wrote the manuscript, EB, SSS and DL designed experiments, analyzed data and wrote the manuscript.

# Additional information

## Funding

| Funder | Author |
|---|---|
| National Cancer Institute | Eric Batchelor |
| National Institute of Diabetes, Digestive and Kidney and Kidney Diseases | S. Stoney Simons, Jr. |
| National Cancer Institute | David Levens |

The funders had no role in study design, data collection and interpretation, or the decision to submit the work for publication.

## Author contributions

Zuqin Nie, Conceptualization, Data curation, Software, Formal analysis, Validation, Investigation, Methodology, Resources, Visualization, Writing - review and editing; Chunhua Guo, Conceptualization, Data curation, Formal analysis, Investigation, Methodology, Resources, Validation; Subhendu K Das, Resources, Validation, Investigation, Visualization, Writing - review and editing; Carson C Chow, Conceptualization, Formal analysis, Writing-review and editing; Eric Batchelor, S Stoney Simons Jnr, Conceptualization, Resources, Data curation, Software, Formal analysis, Supervision, Funding acquisition, Investigation, Methodology, Writing - review and editing; David Levens, Conceptualization, Resources, Formal analysis, Supervision, Funding acquisition, Validation, Investigation, Visualization, Methodology, Project administration

## Author ORCIDs

Carson C Chow (iD) https://orcid.org/0000-0003-1463-9553
David Levens (iD) https://orcid.org/0000-0002-7616-922X

## Decision letter and Author response

Decision letter https://doi.org/10.7554/eLife.52483.sa1
Author response https://doi.org/10.7554/eLife.52483.sa2

# Additional files

## Supplementary files

• Supplementary file 1. Sequences of G-blocks (IDT) used to prepare mutant-MB MYC-EGFP plasmids.

• Transparent reporting form

## Data availability

All data generated or analysed during this study are included in the manuscript and supporting files.

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

## Appendix 1

Here, we briefly summarize the kinetic model of gene transcription previously developed and validated (*Blackford et al., 2014*; *Blackford et al., 2012*; *Chow et al., 2015a*; *Chow et al., 2011*; *Chow et al., 2015b*; *Chow and Simons, 2018*; *Dougherty et al., 2012*; *Luo et al., 2013*; *Ong et al., 2010*; *Pradhan et al., 2016*; *Zhang et al., 2013*; *Zhu et al., 2014*). The model considers gene transcription as a sequence of complex forming reactions

$$S + X_1 \Leftrightarrow Y_1$$
$$Y_1 + X_2 \Leftrightarrow Y_2$$
$$\vdots$$
$$Y_{cls-1} + X_{cls} \Leftrightarrow Y_{cls}$$
$$Y_{cls} + X_{cls+1} \Leftrightarrow Y_{cls+1} \rightarrow P$$
$$\vdots$$
$$Y_{n-1} + X_n \Leftrightarrow Y_n \rightarrow P$$

where $S$ is a steroid agonist, $P$ is the final protein product, $X$'s are factors involved in transcription (called accelerators), and $Y$'s are intermediate complexes formed during transcription. The first reaction represents the binding of steroid to steroid receptor, which is followed by subsequent reactions including the binding of an activated receptor-steroid complex to the hormone response element on the DNA and the recruitment of transcription factors that eventually lead to the gene product. The dose response is the amount or concentration of P as a function of the concentration of S. Experimentally, the dose response is generally found to obey a Michaelis-Menten (MM) form (Hill coefficient of 1), indicating non-cooperativity, which is not expected for such a reaction sequence. However, as shown previously (*Blackford et al., 2014*; *Blackford et al., 2012*; *Chow et al., 2015a*; *Chow et al., 2011*; *Chow et al., 2015b*; *Chow and Simons, 2018*; *Dougherty et al., 2012*; *Luo et al., 2013*; *Ong et al., 2010*; *Pradhan et al., 2016*; *Zhang et al., 2013*; *Zhu et al., 2014*), a MM form is possible if factors X are limited before and at a concentration limiting step (CLS) and abundant after the CLS, with respect to their binding affinity. Previous experiments have found that the binding-to-DNA reaction is the CLS step, as expected since the DNA binding site is generally the most limited quantity in steroid-induced transcription. The dose-response curve for the product of each reaction after the CLS has the identical shape. They can thus be summed with an arbitrary positive weight and still maintain the MM dose-response curve. The biochemical implication is that following the addition of each accelerator to the assembled complex on the DNA there is the possibility of producing the final gene product or adding another accelerator.

Each reaction in the above scheme can also incorporate the effects of inhibition with the additional reactions

$$Y_{i-1} + X_i \; \underset{}{\overset{q_i}{\rightleftharpoons}} \; Y_i^* \to Y_i$$
$$+ \qquad\qquad +$$
$$D_i \qquad\qquad D_i$$
$$\gamma q_i' \updownarrow \qquad\qquad \updownarrow \alpha q_i'$$
$$Y_{i-1} + X_i' \; \underset{(\alpha/\gamma)q_i'}{\overrightarrow{\phantom{aaa}}} \; Y_i' \underset{\beta}{\to} Y_i$$

where $D$ is the inhibitor or decelerator and $q$, $q'$ are binding affinity constants. The decelerator diverts the factor or the new complex away from the reaction between the accelerator and the complex from the previous step. Using the terminology of enzyme kinetics, $\alpha = 0$ indicates *competitive* inhibition, $\gamma = 0$ is *uncompetitive* inhibition, $\alpha = \gamma$ is *noncompetitive* inhibition, $\beta = 0$ is *linear* inhibition, and $\beta > 0$ is *partial* inhibition. Within this framework, a partial inhibitor diverts the step to an alternative pathway, and thus acts like an accelerator. Given the ambiguity of the net effect of a factor we adopted the terms accelerator and decelerator for $X$ and $D$ respectively. Note that $X$ represents the concentration of the active form of the accelerator. For example, $X$ could represent the MYC-MAX dimer, if this was the necessary active form required for MYC action. If the relationship between active MYC and added MYC is linear then active MYC concentration will be proportional to added MYC concentration.

As shown before, the dose-response curve can be completely solved for an arbitrary number of reactions and has the form

$$[P] = \frac{\Gamma V_1^{cls}[S]}{1 + [S]}$$

where

$$\Gamma = \sum_{k=cls}^{n} a_{k-cls} V_{cls+1}^k, \; V_b^m = \prod_{i=b}^{m} v_i, \; W_b^m = \sum_{i=b}^{m} w_i V_b^{i-1}$$

for positive weights and variables

$$v_i = \frac{q_i[X_i](1 + \alpha_i \beta_i q_i[D_i])}{1 + \gamma_i q_i'[D_i]}, \; w_i = \frac{q_i(\epsilon_i + \alpha_i q_i'[D_i])}{1 + \gamma_i q_i'[D_i]}, \; \text{for } i < cls$$

$$v_i = \frac{q_i[X_i](1 + \alpha_i \beta_i q_i[D_i])}{1 + \gamma_i q_i'[D_i]}, \; w_i = \frac{q_{cls}\left( \sum_{k=cls}^{n} \epsilon_k \prod_{j=cls+1}^{k} v_j + \alpha_{cls} q_{cls}'[D_{cls}] \right)}{1 + \gamma_{cl} q_{cls}'[D_{cls}]}, \; \text{for } i = cls$$

$$v_i = \frac{q_i[X_i]}{1 + \gamma_i q_i'[I_i]'}, \; w_i = 0, \; \text{for } i > cls$$

Thus, each factor appears at most once in both formulas. In general, we do not know how many factors participate in gene transcription but if we consider a small number of factors in a given experiment, we can fold the action of the unknown factors into effective coefficients, thereby isolating the action of the factors of interest. As seen above, the action of a given factor depends on its type (e.g. accelerator, competitive decelerator, etc.) and whether it acts before, at, or after the CLS. For example, an accelerator $X$ will affect $A_{max}/EC_{50}$ and $1/EC_{50}$ differently depending on whether it acts before, at, or after the CLS. The predicted forms will be $A_{max}/EC_{50} = B_1[X]$, $1/EC_{50} = B_2 + B_3[X]$, for $X$ before the CLS, $A_{max}/EC_{50} = B_1[X]$, $1/EC_{50} = B_2$, for $X$ at the CLS, and $A_{max}/EC_{50} = B_0 + B_1[X]$, $1/EC_{50} = B_2 + B_3$, for $X$ after the CLS, where square brackets indicate concentration; all the coefficients are positive and do not have the same values for each case. Importantly, the B coefficients are all increasing functions of accelerator binding affinity.

However, there is a second possibility for which the original theory had not considered, which is that factors can act on other accelerators while still preserving the MM dose-response function (with respect to Dex). This is because, as seen in the dose-response formula, the functional form of accelerators and decelerators can be changed without affecting the MM form of the dose-response function. For example, if MYC were to interact with an accelerator $X_i$ in a series of complex forming reactions c.f.

$$X_i + MYC \Leftrightarrow XMYC$$
$$XMYC + MYC \Leftrightarrow X2MYC$$

and free and bound forms of are all viable then we can replace in the dose-response formula with

$$q_i X_i = q_i(c_0 + c_1[MYC] + c_2[MYC]^2)X_i$$

where the c's are positive coefficients. This would make a quadratic function of [MYC]. If $X_i$ acted at the CLS then this would automatically ensure that $1/EC_{50}$ has no dependence on MYC. If it acts after the CLS then the binding affinity of the (unknown) accelerator that MYC facilitates is weak compared to nearby post-CLS accelerators. In this scenario, MYC acts as a *facilitator* of an accelerator and its action is conveyed through this reaction rather than its direct binding to DNA. This other accelerator could be MAX. For example, the MYC-MAX complex accelerates transcription.

F4 has been previously characterized as an inhibitor of MYC-MAX dimerization. However, in our experiments, F4 was found to decrease in the absence of MYC. This implies that the action of F4 extends beyond interfering MYC-MAX dimerization. For example, if F4 were to also block MYCN dimerization, which also activates the CLS accelerator, then the combined action of F4 as a competitive inhibitor with MYC and MYCN could possibly take the form

$$A_{max}/EC_{50} = \left( B_0 + B_1 \frac{[MYC]}{1 + q'[F4]} + B_2 \frac{[MYC]^2}{1 + q'[F4]^2} \right) \frac{1}{1 + d_1[F4] + d_2[F4]^2}$$

In which case would be a supra-linear function of F4, as seen in *Figure 8—figure supplement 1*.

