## [Decision Letter]

Thank you for submitting your article "Dissecting transcriptional amplification by MYC" for consideration by *eLife*. Your article has been reviewed by three peer reviewers, one of whom is a member of our Board of Reviewing Editors, and the evaluation has been overseen by Kevin Struhl as the Senior Editor. The following individual involved in review of your submission has agreed to reveal their identity: Charles Y Lin.

The reviewers have discussed the reviews with one another and the Reviewing Editor has drafted this decision to help you prepare a revised submission.

Summary:

This manuscript is an interesting follow-up of a landmark paper (Nie, 2012) from the same laboratory demonstrating that MYC is required for induction of global mRNA production upon mitogenic stimulation of primary B-cells , revealing that MYC can act as a global transcription enhancer. The current manuscript extends these findings by showing that ectopic expression of MYC enhances transcription from a promoter that is pre-activated by a glucocorticoid receptor, but much less so a basal promoter, suggesting that MYC "amplifies" transcription in this setting. The authors go on to map domains of MYC that either positively nor negatively affect the ability to amplify transcription. A kinetic analysis suggests that MYC acts at more than one stage of the transcription cycle.

Essential revisions:

1) One issue with this manuscript is that a recent analysis of the same experimental system (Sabo et al., 2019) challenges some of the central conclusions of the Nie, 2012 paper and argues that the global MYC-dependent enhancement of transcription is a late and indirect consequence of specific effects of MYC on individual genes. Furthermore, work in P493-6 cells, a model B-cell line, in which MYC levels can be regulated over a large concentration range, has concluded that amplification is a weak and serum batch-dependent effect (Lewis, 2018). While these are simply different data and viewpoints in an ongoing and unresolved scientific debate, these studies raise the possibility that global effects of MYC on gene expression have been overestimated. Therefore, the authors need to show in a physiological setting – such as primary B-cells – that induction of MYC causes a global increase in mRNA synthesis and total mRNA levels ("amplification") and that the MycBox deletion mutants have the expected effects.

2) Given expansive prior data on MYC-MAX binding to GC rich regions and non-canonical e-boxes, the authors should provide schematics fully characterizing the reporters used including the presence of all canonical and non-canonical E-boxes, and the GC% and C+G% of regions.

3) One of the very interesting observations is that deletion of MycBoxI blocks over-amplification by MYC alleles with a deletion of MycBoxes III and IV. What is the explanation for this?

4) The idea that MYC is a general amplifier is closely linked to the notion that it causes global pause-release of RNAPII. The concept put forward here is that MYC must regulate multiple steps in the transcription cycle to amplify transcription. The effects of MYC and the deletion mutants on RNAPII function therefore need to be documented and the authors should clarify whether the effects on reporter genes and the effects of individual deletion mutants can be correlated to distinct steps in the transcription cycle.

5) The observation that the interaction of MYC with promoters depends on the presence of other activators is very interesting. There is no clear mechanism that can explain this observation. So how does the deletion of individual MYCBoxes affect binding? Do MYCBoxes III and IV limit MYC binding to promoters? This seems somewhat at odds with the description of WDR5.

6) The results presented in Figure 1C are interesting and suggest that Myc can cause transcription activation (by a factor of 3) of a non-Ebox promoter upon co-stimulation. However, it is also possible that this is an indirect effect based on more efficient luciferase translation or changes in the cellular metabolism. The authors therefore should analyse if luciferase mRNA levels are regulated as well and if Myc binding to the reporter (reporter ChIP) follows the luciferase signal at different amounts of transfected MYC.

7) There is a huge effect of Dex in Figure 1C but not in Figure3B. No error bars in Figure 3B-H, with the exception of Figure 3F.

8) Strikingly, in transient transfection assays, MYC-∆MBIII behaves like a super-amplifier. Why is this behaviour not evident for stably integrated reporters (5D)? Why is MBIII mutant shown twice in 5D?

9) The ChIP data in Figure 6 is not nicely presented (axes are not labelled, at all).

10) The two-factor competition assay is not described in a way that I can understand the experiment, neither in the legend, nor in the Materials and methods section.

11) Instead of "amplification" the authors should use the neutral term "(reporter) gene activation", whenever possible. E.g. "Driving minimal promoters with exogenous (glucocorticoid receptor) or synthetic transcription factors increased amplification by MYC" could be changed to „Driving minimal promoters with exogenous (glucocorticoid receptor) or synthetic transcription factors increased reporter gene activation by MYC".

12) The Discussion section is too long and the topics extend beyond the scope of this article and should be shortened. Overstatements should be avoided. E.g. „These results unambiguously display the transcription amplification driven by MYC".

---

## [Author Response]

Essential revisions:1) One issue with this manuscript is that a recent analysis of the same experimental system (Sabo et al., 2019) challenges some of the central conclusions of the Nie, 2012 paper and argues that the global MYC-dependent enhancement of transcription is a late and indirect consequence of specific effects of MYC on individual genes. Furthermore, work in P493-6 cells, a model B-cell line, in which MYC levels can be regulated over a large concentration range, has concluded that amplification is a weak and serum batch-dependent effect (Lewis, 2018). While these are simply different data and viewpoints in an ongoing and unresolved scientific debate, these studies raise the possibility that global effects of MYC on gene expression have been overestimated. Therefore, the authors need to show in a physiological setting – such as primary B-cells – that induction of MYC causes a global increase in mRNA synthesis and total mRNA levels ("amplification") and that the MycBox deletion mutants have the expected effects.

The reviewers have asked us to address criticisms of the amplifier hypothesis arising in the papers of Tesi et al., 2019 and Lewis et al., 2018. We address these misguided criticisms below:

a) Tesi et al., (Sabo, above) claim that the increase in bulk RNA occurs only late (24 hours) after MYC activation, and so conclude that the accumulation of total RNA is an indirect consequence of early mitogenic programs triggered by MYC. This paper (Tesi et al.,) is too conceptually and technically flawed to justify re-doing many of the experiments we reported in our 2012 paper (Nie et al., 2012). They have misananalyzed their data; the conclusions concerning the down- or upregulation of specific MYC targets and the inability to document an increase in total RNA are derived from data analyzed with the algorithm DESEQ-2. This program performs very stringent normalization that wipes-out differences in global expression levels. DESEQ-2 uses median-normalization that will artifactually shift the expression profile of one sample with respect to another when their median expression levels differ by a non-trivial amount. DESEQ-2 also uses a Bayesian approach to shrink the differences between samples along an expression distribution—it implicitly assumes that the expression curves of the different samples are fundamentally similar with only occasional “differentially expressed” (DE) outliers that deviate from this curve. DESEQ-2 will also suppress expression differences when broad zones of the expression spectrum are differentially up- or downregulated. As cited in our manuscript, a discussion archived at the Bioconductor web-site specifically discourages the use of DESEQ-2 for the evaluation of global changes in expression. Even if we re-do the experiments, we will analyze them differently and unless skeptics of the amplifier hypothesis recognize their misuse of DESEQ-2 (and other algorithms) the controversy will not be quelled. The only way we can assess the global up- or down-regulation of mRNA (versus regulation of subsets of mRNA) is to compare RNA-Seq before and after some sort of acute MYC activation or inactivation. The outcome of such analysis is completely dependent on the choice of statistical algorithms and bioinformatic pipelines used to process the data. The motivation of this manuscript was to use simple, non-genomic methods and conventional “old-school” assays to define the fundamental properties of MYC without arguing about the implicit assumptions of the algorithms and pipelines.

b) In the manuscript by Nie et al., (2012) we demonstrated that naïve murine B-cells increase total RNA as early as 8 hours after activation which is 4 hours after MYC peaks. This global increase does not occur without MYC. Because total RNA is 97% rRNA, the global increase in RNA is primarily due to increased rRNA synthesis. Since total RNA and total poly-A RNA move in parallel, total RNA synthesis is a reasonable proxy for mRNA synthesis. Surprisingly, Tesi et al., saw no increase in global RNA transcription until 24 hours post-activation, using an in situ assay. Because of this late rise, they conclude that increased total RNA (and hence rRNA) is an indirect consequence of increased MYC. This is surprising. Ribosomal proteins and rRNA have long been recognized as primary MYC targets from flies (Grewal et al., 2005; Herter et al., 2015) to humans (Grandori et al., 2005; van Riggelen et al., 2010).

At the reviewer’s suggestion we sought a system compatible with *eLife*’s timeframe for revision, that could cleanly switch on MYC activity using MYC-ER/tamoxifen rather than the slower doxycycline induction of MYC expression that might not exclude indirect effects). Therefore we studied HO15.19 MYC-ER cells (O'Connell et al., 2003). These MYC-knockout rat cells stably express MYC-ER. We serum starved cells for 72 hours and then treated with 200 nM tamoxifen. Although these serum-starved cells still contain a large basal amount of RNA, an increase in total RNA was almost statistically significant at 6 hours and became so at 10 and 14 hours after tamoxifen addition (new Figure 1—figure supplement 1; increase in total RNA was ~ 4-fold). Such an experimentally detectable increase in total RNA must largely represent increased rRNA synthesis reflecting ribosome biogenesis. While planning additional experiments to build on this observation, we reviewed the literature and discovered an experiment from the Dirk Eick lab that pulse-labeled HO15.19 MYC-ER cells that showed an immediate 5-fold increase in rRNA transcription upon addition of tamoxifen (Schlosser et al., 2003). They also documented an immediate increase in the expression of nucleolar proteins and other components required for ribosome biogenesis. In parallel they demonstrated the same immediate increase in rRNA synthesis and nucleolar protein mRNAs in P493-6 B-cells upon MYC activation. Together with other published work from the labs of Ingrid Grummt (Arabi et al., 2005) with Anthony Wright, it is clear that ribosome biogenesis is an immediate early target of MYC activation and hence the increase in rRNA, and consequently total cellular RNA, commences early. Due to the other exigencies of manuscript revision and because we feel that we cannot improve on the existing literature, we now include our HO15.19 data in Figure 1—figure supplement 1 and cite these earlier studies to show that global RNA levels are rapidly increased by MYC.

c) The reviewers also refer to the Replication Study by Lewis et al., of transcription amplification by MYC in P493-6 cells. This study strongly confirms the MYC amplification from the Rick Young lab (Lin et al., 2012). I encourage the editors and reviewers to re-read that paper along with the commentary by Eick, 2018. Also, please see the “Annotations” appended to this study, one by Rick Young, and the second by one of the current authors, D. Levens. One issue raised by Lewis is rather trivial and simply relates to where to set the arbitrary threshold separating silent and expressed genes; there is no fundamental disagreement on the results. The second issue raised by Lewis et al., relates to variability between lots of serum in their hands—as noted in the annotation by Levens, the failure of Lewis to use charcoal-stripped serum, leaves the estrogen level of the media uncontrolled. Estrogen will activate an EBNA2-ER fusion resident in P493-6 cells and allow these cells to grow in the absence of MYC-induction. This oversight leaves the replication study potentially compromised in this regard.

2) Given expansive prior data on MYC-MAX binding to GC rich regions and non-canonical e-boxes, the authors should provide schematics fully characterizing the reporters used including the presence of all canonical and non-canonical E-boxes, and the GC% and C+G% of regions.

We have now supplied the requested information in the new Figure 1—figure supplement 2. It should be noted that canonical E-boxes CACGTG occur rarely. The only ones that occur in our transfected reporters were engineered into the promoter by us. In the lentivirus, there are three canonical E-boxes, one in each LTR, and one engineered into the reporter promoter by us. It should be noted that, in a 50% G-C sequence, a 5/6 E-box will occur by chance once every 228 bp. So, the large number of 5/6 matches is just as expected. We have no evidence that any of these are functional. Our earlier work (Nie et al., 2012) indicates that E-boxes must be situated close to the transcription start-site to amplify transcription.

3) One of the very interesting observations is that deletion of MycBoxI blocks over-amplification by MYC alleles with a deletion of MycBoxes III and IV. What is the explanation for this?

We agree; this is indeed a very interesting observation that may provide a strong clue as to the mechanism of MYC transactivation. In out first version we stated, “Co-mutation of MBI and MBIII reduced the excessive gain of the MBIII mutation back to wild-type levels (Old Figure 4E now Figure 5E) inviting consideration of MBI as a regulator of mutant MBIII activity.” and did not conjecture further. In the current manuscript we elaborate a bit further: “If MYC participates at multiple points in the transcription-cycle, perhaps by ferrying in different components, then cyclic MBI/phosphorylated T58-directed MYC degradation may help to ratchet the transcription machinery through various stages culminating in pause-release and efficient elongation”. Basically, the degradation of MYC would provide an energetic impulse driving the reaction forward. These concepts build on our observations as well as observations from the Tansey and Sears groups. We now comment on this in the Results section and in the Discussion section.

4) The idea that MYC is a general amplifier is closely linked to the notion that it causes global pause-release of RNAPII. The concept put forward here is that MYC must regulate multiple steps in the transcription cycle to amplify transcription. The effects of MYC and the deletion mutants on RNAPII function therefore need to be documented and the authors should clarify whether the effects on reporter genes and the effects of individual deletion mutants can be correlated to distinct steps in the transcription cycle.

The reviewers raise an excellent point; however a complete description relating MYC’s biochemical roles in the transcription cycle with the formal kinetic steps that we mathematically defined, we believe to be not only beyond the scope of this manuscript, but technically very difficult if not presently impossible. We explicitly investigate a simple transfected reporter gene vs. endogenous genes, but also we make no claim to be able to identify the precise biochemical steps at which MYC acts. The power of the mathematical analysis lies in its ability to constrain biochemical models and guide future experiments. The fact that we now have evidence that MYC acts at more than the one step cited by the reviewer is by itself a significant advance in understanding MYC action.

5) The observation that the interaction of MYC with promoters depends on the presence of other activators is very interesting. There is no clear mechanism that can explain this observation. So how does the deletion of individual MYCBoxes affect binding? Do MYCBoxes III and IV limit MYC binding to promoters? This seems somewhat at odds with the description of WDR5.

There is a straightforward mechanism to explain the recruitment of MYC to activated promoters. As shown in Lorenzin et al., 2015, and as summarized in Wolf et al., 2016, MYC at promoters binds both to DNA and with the transcription and chromatin machineries. If an activator shares a protein partner with MYC and has pre-recruited or co-recruited that partner, then energetically that activator must enhance/stabilize MYC binding.

To address this question further, we explored how MB mutations (alanine-substitutions, not deletions) influence MYC-binding with promoters. Using lentivirus encoded, doxycycline-inducible wild-type or mutant MYCs, as described in the manuscript, we monitored both EZH2 expression and MYC-binding at the EZH2 promoter (new Figure 7). We (and others) find EZH2 to be a reliable reporter of MYC-driven transcription. The results are stunning, and consistent with the observations and hypotheses of the prior version of our manuscript. We find that whereas wild-type MYC (wtMYC) and MBIII-mutant drove high levels of EZH2 expression, MBII- and MBIV-mutants did not. In contrast, binding was most apparent with wtMYC and with MBII-mutant. How can we reconcile these confusing data? If MYC binds to promoters simply by mass-action, and if promoter output simply reflects the equilibrium or steady state levels of promoter-bound MYC, then these results make no sense. But if several cycles of MYC binding and release/degradation/modification recur for each mRNA as MYC kinetically ratchets the transcription machinery through the various stages of the transcription-cycle, leading to productive elongation, then these results are easily rationalized. Mutation of the different MBs impairs orderly progression through the steps of the transcription-cycle. If MBII mutation were effectively a “wrench in the gears” then the mutant MYC might jam the promoter and remain frozen there unable to increase transcription.

In contrast, though mutant-MBIII is found at levels much lower the wild-type protein, it activates transfected reporter genes better than the latter, it activates the native *EZH2* gene almost as well as wtMYC. Yet, by ChIP-qPCR, mutant-MBIII binding to the EZH2 gene was greatly attenuated compared with wtMYC. The simplest way to reconcile these observations is that mutant-MBIII is consumed at a greater rate during the transcription-cycle than is the wild-type protein, and that such consumption is obligatory for MYC action. As MYC turnover is directed by MBI and coupled with T58 phosphorylation and recruitment of E3s such as Fbxw7, co-mutation of MBI would be expected to blunt the enhanced-turnover of mutant MBIII and canceling its hyperactivity. So the hyperactivity of mutant-MBIII is funneled through MBI.

6) The results presented in Figure 1C are interesting and suggest that Myc can cause transcription activation (by a factor of 3) of a non-Ebox promoter upon co-stimulation. However, it is also possible that this is an indirect effect based on more efficient luciferase translation or changes in the cellular metabolism. The authors therefore should analyse if luciferase mRNA levels are regulated as well and if Myc binding to the reporter (reporter ChIP) follows the luciferase signal at different amounts of transfected MYC.

Reporter assays in general and luciferase reporters in particular have been a mainstay of molecular biological studies of promoter activity—including studies of MYC—for more than four decades and for ~37,500 publications. The vast majority of these report effects at the transcriptional level. Because of its protean influence, global transcription activation by MYC may (and perhaps inevitably will) provoke feedback via its influence on translation and metabolism. But there is every reason to expect, based on many hundreds of papers in the MYC literature, that the primary effect is on transcription. In our studies, MYC activates non-E-box promoters exactly in parallel with E-box promoters, the difference being only of degree not of quality. Neverthless, we performed transient transfection experiments to compare activation of E-box and non-E-box reporters by wtMYC and by MBIII-mutant MYC (in new Figure 4C). Again, we find that MBIII-mutant is a more effective activator than wtMYC on both E-box and non-E-box promoters.

To ask whether the MYC-activates non-E-box reporters at the RNA level, RT-qPCR of cells transfected with *GREtkLu*c along with empty vector, wtMYC or MBIII-mutant was performed. The results show that much of the increased output of this E-box-less promoter indeed occurs at the transcriptional level (New Figure 4). RT-qPCR assays of transfected reporters are inherently noisier than are reporter-enzyme assays; whereas the latter measures primarily correctly initiated mRNAs that sponsor proper translation initiation, the former detects any RNA that includes the reporter-specific primer binding sites (luciferase and Renilla in this case). Background transcription extending throughout the plasmid backbone is an old and well known problem for quantifying mRNAs recovered from transfected cells. It is highly likely that some of background expression that we detect from the empty vector reflects such non-specific transcription.

7) There is a huge effect of Dex in Figure 1C but not in Figure3B. No error bars in Figure 3B-H, with the exception of Figure 3F.

The Dex effects in Figure 1C and Figure 3B are identical. The y-axis scale in the old-figure 1C (and 1D) was carried over from an earlier version and should have been replaced. Note that the y-axes scales of Figure 1C and Figure 3B are different to accommodate the full range of MBIII-mutant activity. If the reviewer will zoom on the panels of Figure 3, he/she will see that there are error bars, but they are smaller than the symbols used to plot the data.

8) Strikingly, in transient transfection assays, MYC-∆MBIII behaves like a super-amplifier. Why is this behaviour not evident for stably integrated reporters (5D)? Why is MBIII mutant shown twice in 5D?

The stably-integrated reporters are driven by stably-integrated lentivrus encoded, doxycycline-inducible MYC and MB-mutant MYCs. Comparison of the immunoblots in Figure 1B versus 5D shows that the lentivirus-encoded MYC-EGFP does not reach the highest levels of expression achieved by the transiently transfected MYC-EGFP (note change relative to endogenous MYC). Because of the non-linear amplification by MYC (upward arc in Figure 1C), the lentivirus MYC-EGFP behaves as an amplifier, but not a “super-amplifier”.

I don’t believe that MBIII is shown twice in Figure 5D. The panels include MBIII and the MBI-III double mutant, each with and without doxycycline induction.

9) The ChIP data in Figure 6 is not nicely presented (axes are not labelled, at all).

The figure has been extensively revised and the axes labelled.

10) The two-factor competition assay is not described in a way that I can understand the experiment, neither in the legend, nor in the Materials and methods section.

A better description of the two-factor assay has now been provided in the Materials and methods section.

11) Instead of "amplification" the authors should use the neutral term "(reporter) gene activation", whenever possible. E.g. "Driving minimal promoters with exogenous (glucocorticoid receptor) or synthetic transcription factors increased amplification by MYC" could be changed to „Driving minimal promoters with exogenous (glucocorticoid receptor) or synthetic transcription factors increased reporter gene activation by MYC".

We respectfully disagree. We believe “amplifier” is a better objective descriptor of MYC’s action than “activator”. In these experiments, without glucocorticoid receptor (GR) or Gal4-VP16, MYC does almost nothing. So by itself, in our experiments, MYC is NOT an activator. MYC amplifies activation by these other transcription factors. Perhaps “co-activator” might be appropriate, but coactivation might imply that both MYC and its partner are required for activation—but this is not so—GR and Gal4-VP16 each act solo, but the degree of transcription activation is increased by the presence of MYC. Isn’t “amplify” the appropriate verb to describe this behavior? Nevertheless we have edited the text to try to describe objectively the results of the experiments.

12) The Discussion section is too long and the topics extend beyond the scope of this article and should be shortened. Overstatements should be avoided. E.g. „These results unambiguously display the transcription amplification driven by MYC".

We have edited the Discussion section for brevity, clarity and objectivity.